# Vinpocetine alleviates lung inflammation via macrophage inflammatory protein-1β inhibition in an ovalbumin-induced allergic asthma model

Won Seok Choi, Hyun Sik Kang, Hong Jo Kim, Wang Tae Lee, Uy Dong Sohn*, Ji-Yun Lee[ORCID]*

College of Pharmacy, Chung-Ang University, Seoul, Republic of Korea

* udsohn@cau.ac.kr (UDS); jylee98@cau.ac.kr (JYL)

## Abstract

Asthma is a well-known bronchial disease that causes bronchial inflammation, narrowing of the bronchial tubes, and bronchial mucus secretion, leading to bronchial blockade. In this study, we investigated the association between phosphodiesterase (PDE), specifically PDE1, and asthma using 3-isobutyl-1-methylxanthine (IBMX; a non-specific PDE inhibitor) and vinpocetine (Vinp; a PDE1 inhibitor). Balb/c mice were randomized to five treatment groups: control, ovalbumin (OVA), OVA + IBMX, OVA + Vinp, and OVA + dexamethasone (Dex). All mice were sensitized and challenged with OVA, except for the control group. IBMX, Vinp, or Dex was intraperitoneally administered 1 h before the challenge. Vinp treatment significantly inhibited the increase in airway hyper-responsiveness ($P<0.001$) and reduced the number of inflammatory cells, particularly eosinophils, in the lungs ($P<0.01$). It also ameliorated the damage to the bronchi and alveoli and decreased the OVA-specific IgE levels in serum, an indicator of allergic inflammation increased by OVA ($P<0.05$). Furthermore, the increase in interleukin-13, a known Th2 cytokine, was significantly decreased by Vinp ($P<0.05$), and Vinp regulated the release and mRNA expression of macrophage inflammatory protein-1β (MIP-1β) increased by OVA ($P<0.05$). Taken together, these results suggest that PDE1 is associated with allergic lung inflammation induced by OVA. Thus, PDE1 inhibitors can be a promising therapeutic target for the treatment of asthma.

## Introduction

Asthma is a chronic inflammatory airway disease that leads to coughing, wheezing, and chest tightness [1]. These symptoms are caused by increased mucus secretion, airway hyper-reactivity, and functional and structural changes in lung tissue [2]. Treatment of asthma is usually dependent on corticosteroids and beta$_2$-agonist as a bronchodilator [3]. Some biologic agents, including anti-IgE and anti-interleukin-5 (IL-5), have been developed recently and used in the treatment of severe asthma [4]. However, the specific mechanisms of asthma remain unclear.

**Data Availability Statement:** All relevant data are within the paper and its Supporting Information files.

**Funding:** This research was supported by the National Research Foundation of Korea (NRF) grant funded by the Korea government (Ministry of Science & ICT) (grant number NRF-2018R1C1B6008326).

**Competing interests:** The authors have declared that no competing interests exist.

Although inhaled corticosteroids are the gold-standard therapy used to treat patients with asthma, long-term use of high-dose inhaled corticosteroids can cause adverse effects, such as hypothalamic–pituitary–adrenal axis suppression, reduced bone growth, and increased risk of opportunistic infections [5]. Therefore, there is still a need for the development of new asthma treatment.

Asthma has been associated with T helper cell 2 (Th2)-mediated immunity due to aberrant production of IL-4, IL5, and IL13. In one study, 50% of asthmatic patients showed Th2-related inflammation [6]. Atopic asthma and the genetic predisposition to produce immunoglobulin E (IgE) to common allergens is driven by IL-4-dependent Ig class switching in B cells [7]. Airway eosinophilia depends on both IL-5 and Stat6 signaling [8]. Each cytokine has distinct functional effects in the induction of disease, but IL-13 predominates in its contribution to the pathophysiology in asthma [9]. IL-13 is now thought to be especially critical, as it promotes goblet cell differentiation, mucus production, bronchial hyper-responsiveness, IgE synthesis, and eosinophil recruitment [10].

Eosinophils play a key role in numerous inflammatory diseases, including allergic disorders [11]. In most asthma phenotypes, there are increases in eosinophils in the tissues, blood, and bone marrow and in general, the numbers increase with disease severity [12]. The eosinophil is the central effector cell responsible for ongoing airway inflammation [12, 13]. When activated, eosinophils can release many mediators, such as eosinophil peroxidase, lysozyme, lipid mediators, cytokines, and chemokines, which contribute to airway hyper-reactivity [14–16]. Excessive eosinophil airway infiltration causes serious symptoms, such as severe cough, dyspnea, and hypoxemia [17]. For eosinophil recruitment and activation, chemo-attractants play a key role in determining the increased tissue localization and activation during disease [18]. Macrophage inflammatory protein-1β (MIP-1β) is a well-known chemokine produced by various cells, such as neutrophils, epithelial cells, B cells, T cells, and eosinophils [17]. In a previous study, MIP-1β displayed chemo-attractant activity for murine eosinophils via C-C chemokine receptor type 5 (CCR5) and was shown to be involved in eosinophil recruitment during airway inflammation [15].

Phosphodiesterase (PDE) inhibitors prevent the inactivation of intracellular cyclic adenosine monophosphate (cAMP) and cyclic guanosine monophosphate (cGMP) [19]. Studies have demonstrated that cAMP and cGMP have a role in airway smooth muscle relaxation and down-regulate the airway inflammation and airway remodeling [20, 21]. Consequently, most PDEs are expressed in lung and immune cells [21]. The cAMP-specific PDE family negatively regulates the function of almost all pro-inflammatory and immune cells and exerts widespread anti-inflammatory activity in animal models of asthma. Some PDE inhibitors have been implicated in the treatment of chronic obstructive pulmonary disease and asthma [22]. For example, Roflumilast, a PDE4 inhibitor, is currently being used to treat chronic obstructive pulmonary disease and effectively improves asthma symptoms [10, 23]. Based on this evidence, we hypothesized that targeting PDEs has potential application in asthma treatment.

PDE1, as one of the subtypes in the PDE superfamily, is expressed in pulmonary arterial smooth muscle cells, epithelial cells, fibroblasts, macrophages, and lymphocytes [21, 24, 25]. It is well-known that PDE1 degrades both cAMP and cGMP [26]. Inhibition of cAMP and cGMP degradation can regulate airway inflammation and airway smooth muscle contraction [20]. Although the direct association between PDE1 and asthma remains unclear, recent studies have provided clues to the relationship between allergic lung inflammation and PDE1. PDE1A and PDE1C protein expression was detected in the isolated lung cells of mice, and it was increased in the inflammation state [27, 28]. A previous study also reported that PDE1A inhibition prevented lung fibrosis [29]. Other work demonstrated that PDE1 inhibition may dampen inflammatory responses of microglia in the disease state [26]. Moreover, PDE1B has

been associated with the activation or differentiation of immune cells [30, 31]. It was found to be expressed in T lymphocytes and modulate the allergic response by regulating IL-13 production, which was closely related to allergic lung inflammation [30–32]. In this context, we hypothesized that PDE1 inhibition could down-regulate allergic inflammation.

Vinpocetine (Vinp), a derivative of the alkaloid vincamine, is a PDE1 inhibitor [33] that exerts a neuroprotective effect by relaxing the cerebrovascular system [34]. It is also known to have anti-inflammatory activity by preventing the increase in tumor necrosis factor-alpha (TNF-$\alpha$) [35]. However, the role of Vinp as a PDE1 inhibitor in lung diseases, such as asthma, remains unclear.

It is known that 3-isobutyl-1-methylxanthine (IBMX), a methylated xanthine derivative, acts as a non-competitive selective PDE inhibitor [36]. In order to understand the action of Vinp more completely, we investigated the relationship between asthma and PDE1 by examining the effect of Vinp on asthma in a murine model. We also compared the effects of IBMX and Vinp on ovalbumin (OVA)-induced allergic lung inflammation.

## Materials and methods

### Materials

OVA, grade V from hen egg white (lyophilized powder, ≥98%) and dexamethasone (Dex; catalog number D1756) were purchased from Sigma–Aldrich (St Louis, MO, USA). Aluminum hydroxide (Imject® Alum) was purchased from Thermo Scientific, Rockford, IL, USA. IBMX and Vinp were obtained from Tocris Bioscience, Bristol, UK. Stock solutions of IBMX, Vinp, and Dex were prepared at 3.5 mg/ml in DMSO. Before the intraperitoneal (i.p.) injection to the mice, IBMX, Vinp, and Dex were diluted to 1 mg/ml with normal saline. All chemicals used in this study were of analytical grade.

### Animals

Balb/c male mice (5 weeks old, weighing 20–30 g) were obtained from Samtako Bio Korea, Gyeonggi-do, Republic of Korea, and housed in the mouse facility in the R&D Center of Chung-Ang University (Seoul, Republic of Korea) with sterilized bedding. The air condition was maintained at 24±2˚C with 50±5% humidity, and the light/dark cycle was synchronized to 12:12 h. Pathogen-free food and water were provided. The mice were acclimated for 1 week, and 10 mice per group were randomized to five treatment groups (control, OVA, OVA + IBMX, OVA + Vinp, and OVA + Dex). Overall health was monitored twice a week. After the experiment, mice were euthanized by intramuscular injection with 40 mg/kg Zoletil 50 (125 mg tiletamine and 125 mg zolazepam; Virbac, Carros, France)–10 mg/kg xylazine (Sigma–Aldrich). All experimental procedures complied with the guidelines established by the Institutional Animal Care and Use Committee of Chung-Ang University, and the study design was approved by the appropriate ethics review board (IACUC 2018–00124).

### OVA-induced asthma model

On days 1, 7, and 14, 100 μg OVA and 1 mg of aluminum hydroxide in 200 μl of normal saline were injected (i.p.) into the mice for sensitization. On days 21, 23, 25, 27, 29, and 31, mice were exposed to 5% OVA in normal saline for 30 min using a nebulizer (Aerogen®, Galway, Ireland). The control group was exposed to normal saline. In the drug-treated groups (OVA + IBMX, OVA + Vinp, OVA + Dex), IBMX, Vinp, or Dex was injected (i.p.). 1 h before 5% OVA exposure. The administration dosage was 10 mg/kg (the injection volume was 0.2 ml per

mouse). In the OVA group, 0.2 ml of the vehicle was injected (i.p.) 1 h before 5% OVA exposure (Fig 1A).

## Measurement of airway resistance and tidal volume of lungs

Airway resistance is the resistance of the respiratory tract to airflow during inhalation and exhalation. Tidal volume is the lung volume representing the normal volume of air displaced between normal inhalation and exhalation when extra effort is not applied. Airway resistance

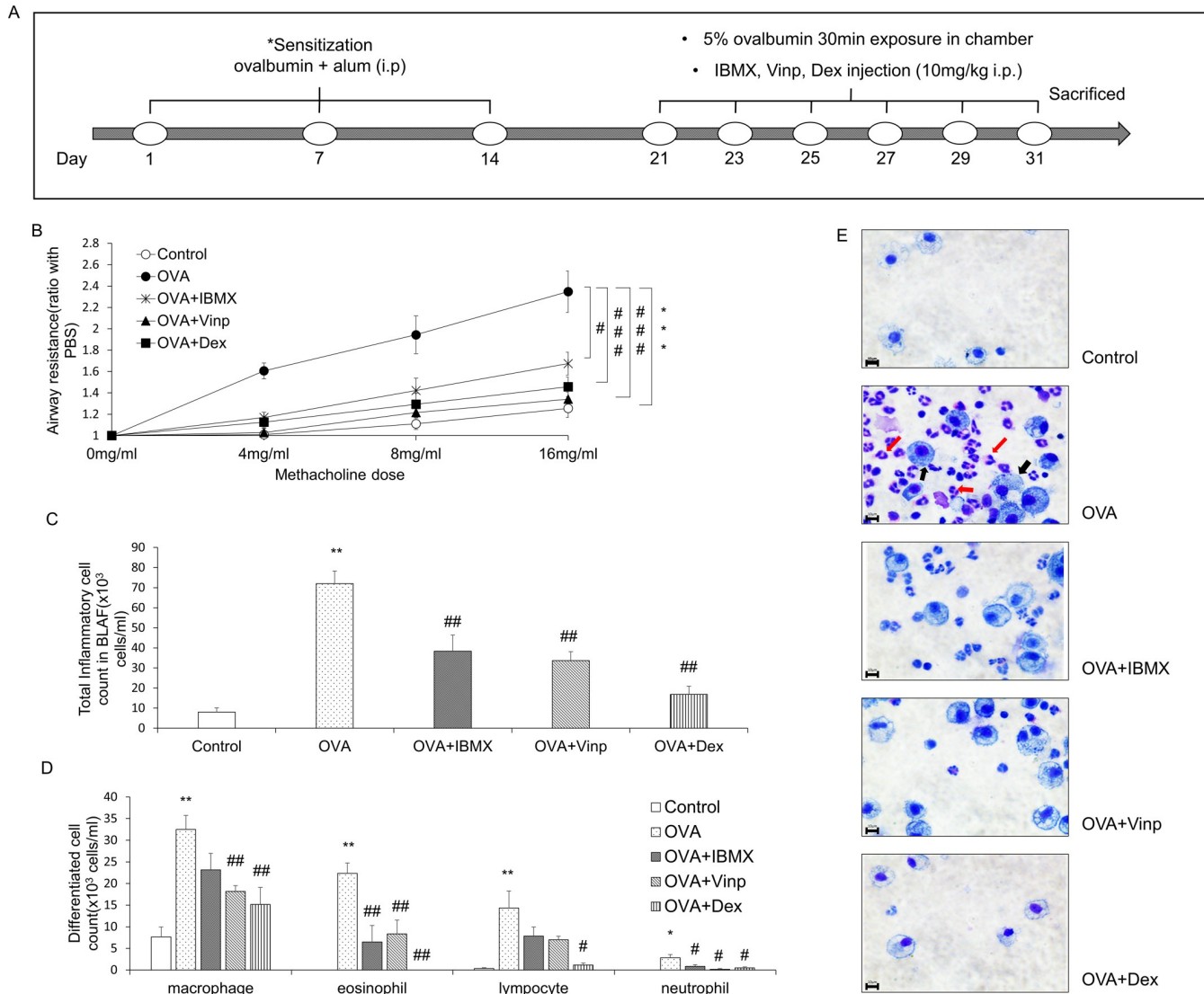

**Fig 1. Effects of IBMX and Vinp in an OVA-induced asthma model.** (**A**) Schedule for inducing asthma in mice model. (**B**) The methacholine test was performed to measure airway resistance. Mice in the OVA, OVA + IBMX, OVA + Vinp, and OVA + Dex groups were exposed to methacholine (4, 8, 16 mg/ml). After methacholine exposure, the airway resistance of the lungs was measured by plethysmography. Inflammatory cells in BALF were stained with Kwik-Diff. (**C**) Stained inflammatory cells were counted by a hemocytometer. (**D**) Differentiated cells (macrophages, eosinophils, lymphocytes, and neutrophils) were analyzed based on standard morphological criteria. (**E**) Stained cells were observed using a microscope (red arrows, eosinophils; black arrows, macrophages; magnification, 63×; scale bar, 10 μm). Data are expressed as mean ± SEM. Statistical analysis was performed using the Student's $t$-test, one-way ANOVA, and two-way ANOVA (Data were considered significant at $^*P<0.05$, $^{**}P<0.01$, and $^{***}P<0.001$ compared with the control group and $^#P<0.05$, $^{##}P<0.01$, and $^{###}P<0.001$ compared with the OVA group). IBMX, 3-isobutyl-1-methylxanthine; BALF, broncho-alveolar lavage fluid; OVA, ovalbumin; Vinp, vinpocetine; Dex, dexamethasone.

and tidal volume can be measured by plethysmography. We also confirmed that the methacholine test did not affect the inflammatory parameter in this study. One day after the final exposure to OVA, mice in the OVA, OVA + IBMX, OVA + Vinp, and OVA + Dex groups were exposed to methacholine (4, 8, and 16 mg/ml) for 1 min. After methacholine exposure, the airway resistance of the lungs was measured for 3 min using a Buxco® non-invasive double-chamber plethysmograph (Data Sciences International, St. Paul, MN, USA). Measured airway resistance and tidal volume were automatically calculated by FinePointe software (Data Sciences International) and expressed as a ratio from the value at 0 mg/ml ($n$ = 9).

## Inflammatory cell-counting in broncho-alveolar lavage fluid (BALF)

After the methacholine test, mice were anesthetized and euthanatized as described above. BALF was obtained by lavage of the right lung, then centrifuged at 1,500$g$ for 10 min. The supernatant was collected and stored at –78˚C for further biochemical analysis (Measurement of cytokine release in BALF). The pellet was suspended in 250 μl PBS, and the suspended cells, representing total inflammatory cells, were counted using a hemocytometer. The differential inflammatory cells, such as eosinophils, macrophages, and neutrophils, were counted based on standard morphological criteria after staining on a glass slide using the Shandon™ Kwik-Diff™ Stain Kit (Thermo Fisher Scientific, Waltham, MA, USA).

## Histological analysis

The left lung was removed from each mouse, fixed with 10% formalin solution, embedded using Tissue-Tek® (Sakura Finetek®, Torrance, CA, USA), then sectioned using a Leica microtome 820 (Leica Microsystems, Wetzlar, Germany) at 4 μm ($n$ = 6). The sectioned tissues were stained with hematoxylin–eosin (H&E). The degree of inflammation in lung tissues was determined by the inflammation scoring system, where 0 = no inflammation, 1 = occasional cuffing with inflammatory cells, 2 = most bronchi or vessels surrounded by a thin layer (1–5 cells) of inflammatory cells, and 3 = most bronchi or vessels surrounded by a thick layer (>5 cells) of inflammatory cells. Periodic acid–Schiff (PAS) staining of the lung tissues was performed using the PAS Stain Kit (Abcam, Cambridge, UK). To quantify mucus production, mucus secretion was evaluated by measuring the broncho-alveolar red-stained regions using the ImageJ software program (NIH Image, Bethesda, MD, USA) [37]. Congo red staining of the lung tissues was performed using the Congo Red Stain Kit (Abcam). Eosinophils were counted in an area of 20,000 μm$^2$ of lung tissues after staining with Congo red [38]. Six random fields of each stained tissue section were observed under a microscope (Leica Microsystems), and images were captured using a Leica DM 480 camera (Leica Microsystems).

## Immunofluorescence staining

MIP-1β protein was detected using 20 μg/ml rabbit anti-CCL4 polyclonal antibody (Invitrogen, Carlsbad, CA, USA, catalog number PA5-34509, lot #UL2898185) and 24 μg/ml goat anti-rabbit IgG FITC secondary antibody (Invitrogen, catalog number 65–6111, lot #UG285467) in the 4-μm paraffin section of mouse lung tissue ($n$ = 6). The nucleus was stained using Ultra-Cruz® Aqueous Mounting Medium with DAPI (Santa Cruz Biotechnology, Inc., Dallas, TX, USA). Six random fields of each immunofluorescence-stained tissue section were observed under a microscope (Leica Microsystems), and images were captured using a Leica DM 480 camera (Leica Microsystems). The fluorescence intensity was measured using the ImageJ software program (NIH Image).

## Measurement of anti-OVA IgE in serum

Blood samples were obtained from the inferior vena cava for biochemical assay ($n = 10$). Obtained blood samples were centrifuged at 1,500$g$ for 10 min, and the serum was separated. Anti-OVA IgE in serum was measured using an ELISA kit (Cayman Chemical, Ann Arbor, MI, USA) according to the manufacturer's instructions. Absorbance was measured using a FlexStation3 Microplate Reader (Molecular Devices, Sunnyvale, CA, USA) at 450 nm according to the manufacturer's protocol. The concentration was determined using a standard curve of manufacturer's IgE standard solution.

## Measurement of cytokine release in BALF

The levels of IL-4, IL-5, IL-13, and MIP-1β in BALF were measured using an ELISA kit (R&D Systems, Inc., Minneapolis, MN, USA) according to the manufacturer's instructions ($n = 6-8$ per group). Absorbance was measured as described above.

## Quantitative analysis of mRNA expression in lung tissues

The level of mRNA expression in the lungs was measured by quantitative reverse transcription PCR (RT-qPCR). The total RNA was extracted from the right lung of each mouse using TRIzol™ reagent (Ambion®, Life Technologies™, Carlsbad, CA, USA). The RNA was spectrophotometrically quantified using a NanoDrop ND-1000 (Thermo Fisher Scientific). One microgram of total RNA was used to synthesize cDNA using the iScript™ cDNA Synthesis Kit (Bio-Rad, Hercules, CA, USA). After synthesis, 100 ng cDNA was used for quantitative PCR (qPCR). qPCR was performed using the iQ™ SYBR® Green Supermix (Bio-Rad). The primer sequences used in this study are shown in Table 1. The CFX96 Real-Time PCR Detection System (Bio-Rad) was used to monitor fluorescent intensity during amplification. cDNA was initially denatured at 95˚C for 3 min, then cycled 40 times through the following cycle: denaturation (95˚C for 10 s), annealing (55˚C for 30 s), plate read. A melting curve was generated after cycling by increasing the temperature from 55 to 95˚C at 0.5˚C for 5 s and performing a plate read at each increment. CFX Manager software was used to automatically calculate the cycle quantification value at which samples amplified at a high enough value to be detected ($\Delta Ct$ values). Each $\Delta Ct$ value was normalized against $GAPDH$, used as a housekeeping gene. Transcript expression was determined relative to the control group. The RT-qPCR analysis was performed in three independent experiments ($n = 6-8$ per group).

## Western blot analysis

Protein expression in the lungs was measured by western blot analysis. Proteins were extracted using RIPA buffer (Thermo Scientific) containing protease inhibitor cocktail (Roche, Basel, Switzerland) and phosphatase inhibitor cocktail (Roche), as per the manufacturer's

**Table 1. The primers used for RT-qPCR in this study.**

| Gene | Forward sequence (5'–3') | Reverse sequence (5'–3') |
|---|---|---|
| IL-13 | AGACCAGACTCCCCTGTGCA | TGGGTCCTGTAGATGGCATTG |
| MIP-1β | CTCAGCCCTGATGCTTCTCAC | AGAGGGGCAGGAAATCTGAAC |
| PDE1A | GAGCACACAGGAACAACAAACA | AAGTCTGTAGGCTGCGCTGA |
| PDE1B | GAGCCAACCTTCTCTGTGCTGA | CGTCCACATCTAAAGAAGGCTGG |
| PDE1C | CAGTCATCCTGCGAAAGCATGG | CCACTTGTGACTGAGCAACCATG |
| GAPDH | CATCACTGCCACCCAGAAGACTG | ATGCCAGTGAGCTTCCCGTTCAG |

instructions. Extracted protein concentrations were quantified by the BCA protein assay reagent (Thermo Scientific) using bovine serum albumin (BSA) as a standard. For electrophoresis, 20 μg of protein was loaded into each well. The protein was separated on a 10% sodium dodecyl sulfate-polyacrylamide gel electrophoresis (SDS-PAGE) gel at 60 V for 30 min, followed by 90 V for 2 h, and then transferred to polyvinylidene fluoride (PVDF) membranes (Merck, Darmstadt, Germany). The membranes were blocked by incubation with 5% BSA in PBS-T buffer (0.1% Tween in PBS, pH 7.6) at room temperature for 1 h to prevent non-specific binding. The membranes were incubated overnight with primary antibodies at 4˚C. Then, the membranes were incubated with secondary antibodies at room temperature for 2 h. The primary antibodies used were all purchased from Santa Cruz Biotechnology, Inc., and included the following: mouse anti-PDE1A antibody (catalog number sc-374602, lot #H1219, dilution ratio 1:200), mouse anti-PDE1B antibody (catalog number sc-393112, lot #E2218, dilution ratio 1:200), mouse anti-PDE1C antibody (catalog number sc-376474, lot #F2717, dilution ratio 1:200), and mouse anti-β-actin antibody (catalog number sc-47778, lot #A2317, dilution ratio 1:1000). HRP-linked horse anti-mouse IgG antibody was used as the secondary antibody (Cell Signaling Technology, Inc., Danvers, MA, USA, catalog number 7076S, lot #33, dilution ratio 1:3000). Quantitative analysis was measured using the ImageJ software (NIH Image). Western blot analysis was performed in three independent experiments ($n = 4-5$ per group).

## Statistical analysis

Values are represented as the mean ± SEM of data from 10 mice ($n = 10$) per group. Data were statistically analyzed by the Student's $t$-test, one-way ANOVA, and two-way ANOVA using GraphPad Prism 7 (GraphPad Software, Inc., San Diego, CA, USA). The significance level was $^*P{<}0.05$ versus the control group and $^#P{<}0.05$ versus the OVA group.

## Results

### Effects of IBMX and Vinp on airway hyper-responsiveness and changes in inflammatory cells in BALF induced by OVA

To measure airway hyper-responsiveness, we performed the methacholine test. An increase in the methacholine dose induced an increase in airway resistance. In the OVA group, airway resistance was significantly escalated ($P{<}0.001$), and this increase was significantly alleviated by treatment with IBMX, the non-specific PDE inhibitor ($P{<}0.05$). Vinp, the PDE1-specific inhibitor, reduced the airway resistance more than IBMX ($P{<}0.001$). Dex, which served as a positive control, significantly inhibited the increase in airway resistance induced by OVA ($P{<}0.001$) (Fig 1B).

The inflammatory cells in BALF were counted to investigate the infiltration of inflammatory cells into the lungs. The total inflammatory cells in BALF increased more in the OVA-exposed group than in the control ($P{<}0.01$). IBMX and Vinp significantly reduced the increase in the total inflammatory cells ($P{<}0.01$). Dex (positive control) also decreased the total inflammatory cells significantly ($P{<}0.01$; Fig 1C). In particular, eosinophils, macrophages, lymphocytes, and neutrophils were increased by OVA exposure ($P{<}0.01$ for all, except for neutrophils, where $P{<}0.05$). IBMX and Vinp reduced these inflammatory cells. In particular, IBMX and Vinp caused a large decrease in eosinophils ($P{<}0.01$; Fig 1D and 1E).

### Effects of IBMX and Vinp on inflammatory cell infiltration in lung tissue

The histological changes in each group were monitored by H&E staining (Fig 2A and 2D). The alveoli and bronchi were thicker in the OVA group than in the control group. The OVA group also showed the infiltration of inflammatory cells into the lungs ($P{<}0.001$). IBMX ameliorated

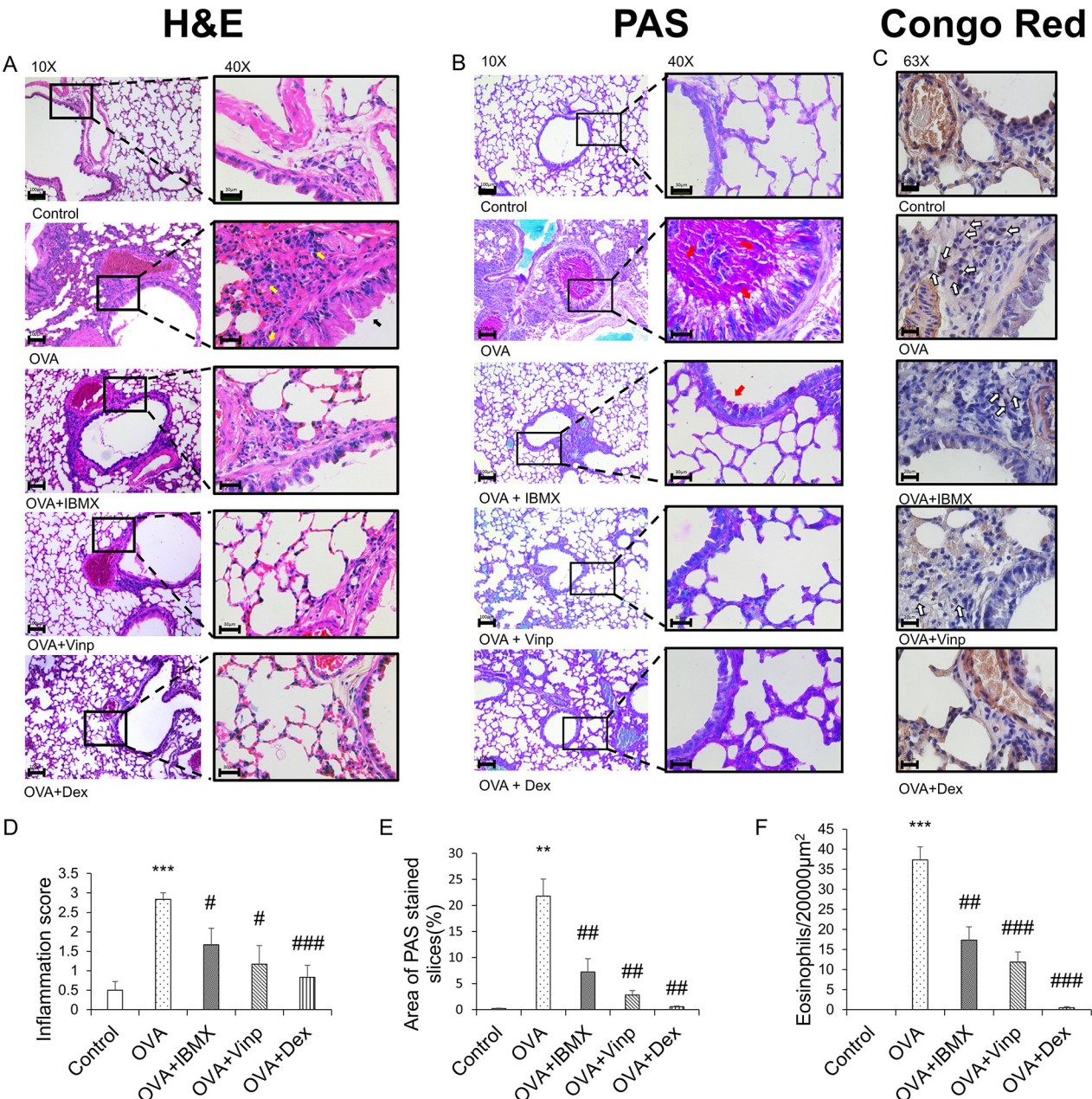

**Fig 2. Inhibition of ovalbumin-induced histological changes by IBMX and Vinp.** (**A**) H&E, (**B**) PAS, and (**C**) Congo red staining were performed to investigate the histological changes. H&E and PAS staining (left-hand side: magnification 10×; scale bar, 100 µm; right-hand side: magnification 40×; scale bar, 30 µm) show infiltration of inflammatory cells (yellow arrows), damage to epithelial cells (black arrows), and mucus stained magenta (red arrows). Congo red staining (magnification 63×, scale bar, 20 µm) shows the eosinophils (white arrows). (**D**) The inflammation scores were determined based on criteria. (**E**) PAS-stained areas were analyzed using ImageJ software. (**F**) Congo red-stained eosinophils were counted in a 20,000-µm² area. Data are expressed as mean ± SEM. Statistical analysis was performed using one-way ANOVA (Data were considered significant at **$P<0.01$ and ***$P<0.001$ compared with the control group and #$P<0.05$, ##$P<0.01$, and ###$P<0.001$ compared with the OVA group). IBMX, 3-isobutyl-1-methylxanthine; H&E, hematoxylin–eosin; PAS, periodic acid–Schiff; OVA, ovalbumin; Vinp, vinpocetine; Dex, dexamethasone.

these effects slightly ($P<0.05$). Vinp also improved the morphological abnormality induced by OVA in the lungs ($P<0.05$). Mucus secretion (Fig 2B and 2E) and eosinophil recruitment in lung tissues (Fig 2C and 2F) were increased in the OVA group ($P<0.01$). IBMX and Vinp treatment reduced the mucus secretion compared with the OVA group ($P<0.01$) and

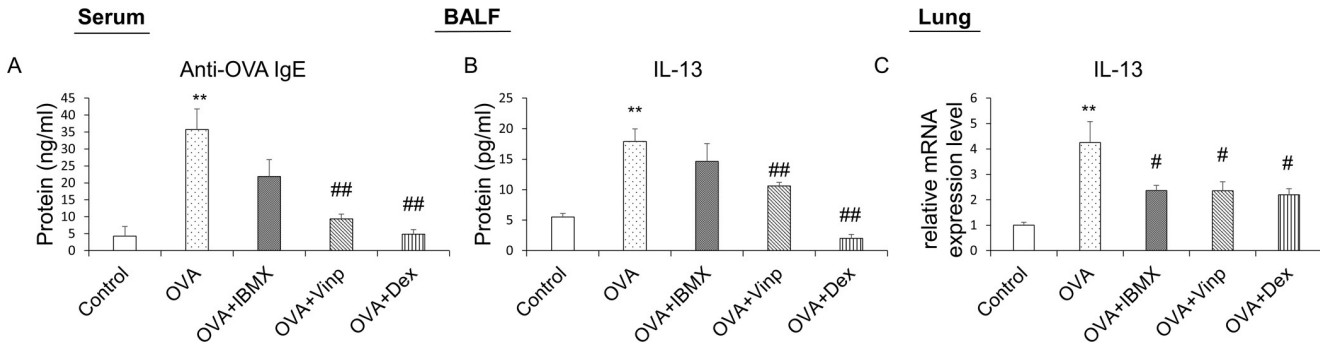

**Fig 3. Effects of IBMX and Vinp on OVA-induced increase in allergic inflammatory mediators.** Plasma samples obtained from mice were analyzed by ELISA to investigate systemic allergic inflammation. (**A**) Anti-OVA IgE in serum and (**B**) the release of IL-13 in BALF were measured by ELISA. The mRNA expression levels of (**C**) IL-13 in lung tissues were measured by RT-qPCR. Data are expressed as mean ± SEM. Statistical analysis was performed using one-way ANOVA (Data were considered significant at $^{**}P<0.01$ compared with the control group and $^{#}P<0.05$ and $^{##}P<0.01$ compared with the OVA group). IBMX, 3-isobutyl-1-methylxanthine; BALF, broncho-alveolar lavage fluid; OVA, ovalbumin; Vinp, vinpocetine; Dex, dexamethasone.

decreased the infiltration of eosinophils significantly (IBMX; $P<0.01$, Vinp; $P<0.001$). Dex (positive control) significantly decreased these histological changes (H&E staining; $P<0.001$, PAS staining; $P<0.01$, Congo red staining; $P<0.001$).

## Down-regulation of allergic inflammation by IBMX and Vinp through OVA-induced decrease in IgE and IL-13

OVA-specific IgE was analyzed by ELISA to measure the degree of systemic allergic inflammation. OVA-specific IgE was increased significantly in the OVA-exposed group ($P<0.01$). In the Vinp group, the OVA-specific IgE level in serum was reduced significantly compared with the OVA group ($P<0.01$). Dex (positive control) also reduced the IgE levels ($P<0.01$; Fig 3A).

IL-13 in BALF was analyzed to investigate the release of inflammatory cytokines. The OVA group showed a significant increase in the release of IL-13 ($P<0.01$). Although IBMX did not reduce the release of IL-13, Vinp inhibited the release of IL-13 significantly ($P<0.01$). The Dex group served as the positive control ($P<0.01$; Fig 3B).

We performed RT-qPCR using mouse lung tissues to analyze the mRNA expression levels in the lungs associated with allergic inflammation. The IL-13 expression level in the lungs was increased significantly in the OVA group ($P<0.01$). The increase in IL-13 expression was down-regulated by IBMX, Vinp, and Dex ($P<0.05$; Fig 3C).

## OVA-induced increase in MIP-1β inhibited by both IBMX and Vinp

MIP-1β is a chemokine involved in a variety of inflammatory responses. To investigate the changes in MIP-1β, we performed ELISA, RT-qPCR, and immunofluorescence staining. OVA significantly induced MIP-1β release in BALF ($P<0.01$) and mRNA expression in the lungs ($P<0.05$). IBMX and Vinp both down-regulated the release ($P<0.01$) and mRNA expression of MIP-1β ($P<0.05$). Dex significantly reduced the MIP-1β release in BALF ($P<0.01$), but the MIP-1β mRNA expression level did not reduce significantly compared with the OVA group (Fig 4A and 4B). Immunofluorescence staining supported the MIP-1β expression results in the lungs (Fig 4C and 4D).

## Up-regulation of PDE 1 expression in the lungs of OVA-induced asthma mice model

To investigate the changes in the expression of PDE1A, 1B, and 1C levels in the lungs following OVA exposure, we performed western blot and RT-qPCR analysis. For all three proteins, the

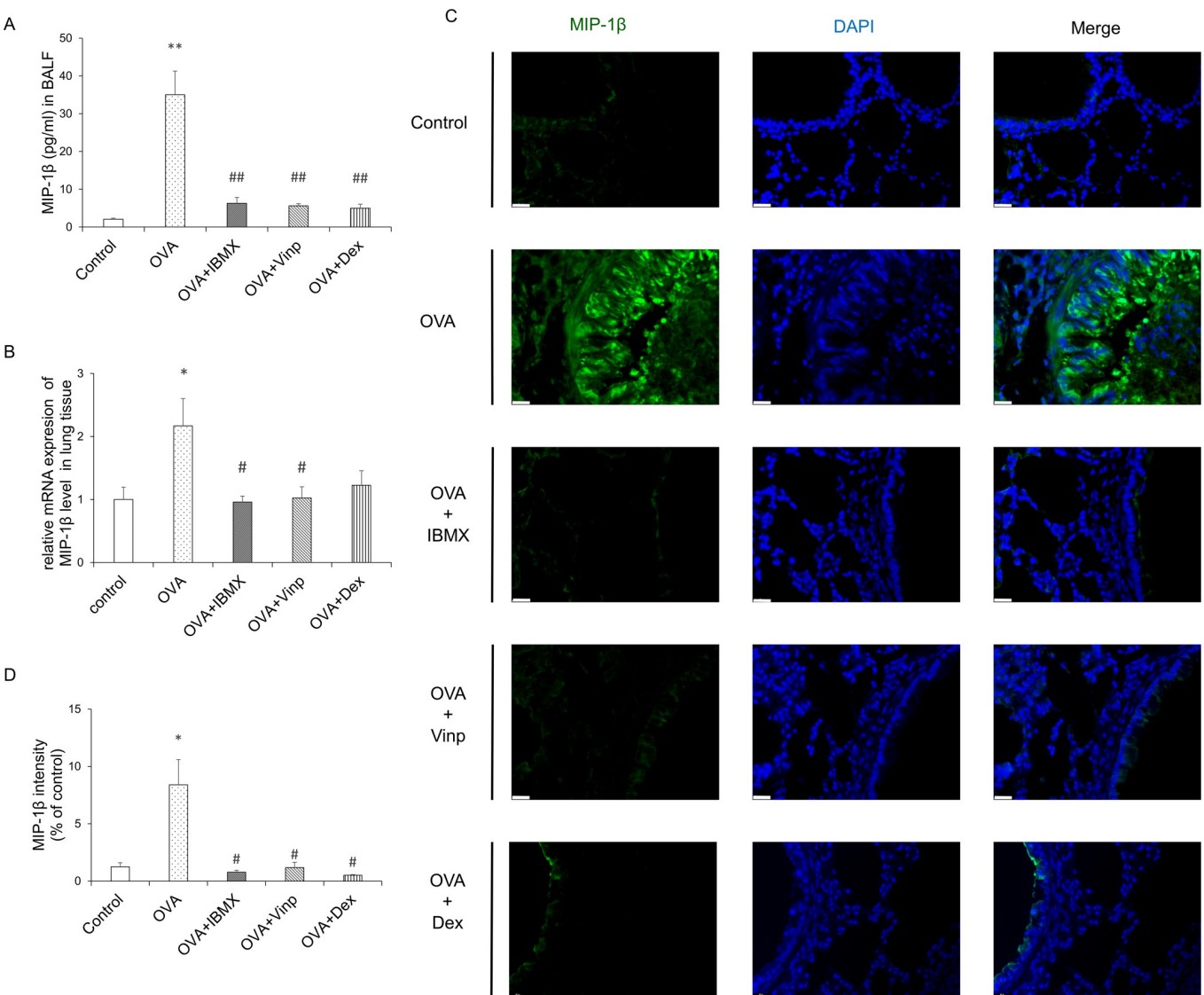

**Fig 4. Effects of IBMX and Vinp on MIP-1β expression and release in eosinophilic lung inflammation.** (**A**) MIP-1β in BALF was measured by ELISA. (**B**) The mRNA expression level of MIP-1β in lung tissues was measured by RT-qPCR. (**C**) MIP-1β expression in lung tissue was detected by immunofluorescence staining (magnification, 63×; scale bar, 20 μm). (**D**) In immunofluorescence-stained tissue, the green fluorescence intensity of stained images was measured using ImageJ. Data are expressed as mean ± SEM. Statistical analysis was performed using one-way ANOVA (Data were considered significant at $^{**}P<0.01$ compared with the control group and $^{#}P<0.05$ compared with the OVA group). IBMX, 3-isobutyl-1-methylxanthine; BALF, broncho-alveolar lavage fluid; Vinp, vinpocetine; MIP-1β, macrophage inflammatory protein-1β; OVA, ovalbumin; Dex, dexamethasone.

expression levels increased significantly in the OVA group relative to the control ($P<0.05$ for all, except PDE1C, where $P<0.01$; Fig 5A). The mRNA expression levels of PDE1A, 1B, and 1C were also significantly up-regulated in the OVA group ($P<0.05$ for all; Fig 5B).

## Discussion

It was known that PDE inhibitors have effect on respiratory disease such as asthma and chronic obstruction pulmonary disease [22, 39]. In particular, PDE 3, 4, and 7 inhibitors are known to be effective in mouse models of asthma from previous studies [22]. However, studies on asthma using the PDE 1 inhibitor in allergic asthma mice model have not been studied yet.

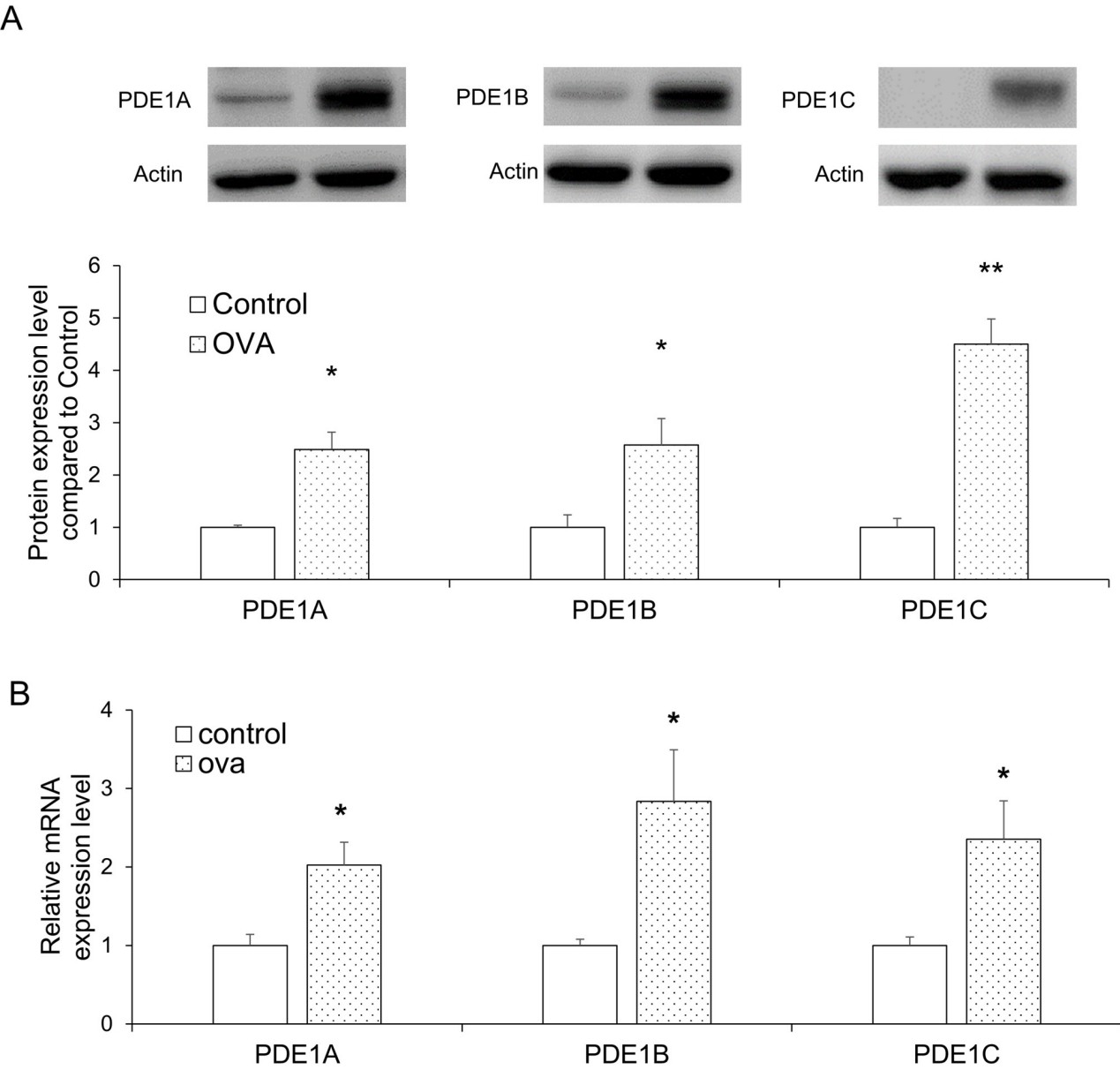

**Fig 5. PDE1 expression levels in lung tissues were selectively increased by OVA exposure.** (**A**) PDE1A, PDE1B, and PDE1C protein expression levels in the lung tissues were analyzed by western blot. (**B**) PDE1A, PDE1B, and PDE1C mRNA expression levels in the lung tissues were measured by RT-qPCR. Data are expressed as mean ± SEM. Statistical analysis was performed using the Student's $t$-test (*$P<0.05$ and **$P<0.01$ were compared with the control). OVA; ovalbumin.

In this study, we investigated whether Vinp, a known PDE1 inhibitor, affects the OVA-induced asthma model. This study also focused on the contribution of PDE1 to asthma and its potential as a therapeutic target. To investigate the relationship with asthma, the appropriate asthma mice model and treatment schedule were determined from previous research and a pilot study [5, 10, 40–42]. In the asthma model, OVA exposure led to an increase in airway sensitivity. However, both IBMX and Vinp treatment, respectively, significantly inhibited the increased airway resistance. Furthermore, Vinp treatment alleviated the increased airway resistance more than the IBMX treatment. Considering that IBMX is a non-selective PDE inhibitor

and that Vinp is a selective PDE1 inhibitor, these results indicate that PDE, especially PDE1, is associated with airway hypersensitivity. Previous research reported that PDE4 inhibition was shown to have an effect on the attenuation of airway resistance in asthma mice model [43]. PDE3 inhibitor and PDE1/4 dual inhibitor, has been reported to cause the inhibition of the ovalbumin-induced bronchoconstriction [44]. These previous studies supported that PDE inhibition can alleviated the increased airway resistance. Through this study, we showed that PDE 1 inhibitor can ameliorated the airway resistance increased in asthma mice model. In tidal volume, there were no significant differences among the groups (S1 Fig). The mouse model of asthma induced in this study does not appear to be associated with a decrease in tidal volume.

Patients with asthma have hypersensitive airways because the inflammatory cells infiltrate the airway and lungs [45]. The infiltrating inflammatory cells release cytokines, a major factor in airway hyper-responsiveness [10]. Among the inflammatory cells, eosinophils are well-known to increase airway hyper-responsiveness [46]. It was reported that PDE 4 inhibitor reduced the number of eosinophils, neutrophils, and lymphocytes in BALF [47]. In this study, OVA exposure increased the total inflammatory cells in BALF. In particular, the number of eosinophils increased significantly. This increase was significantly inhibited by the administration of both IBMX and Vinp, individually. As a result, it showed that PDE inhibition amelio-rated eosinophilia in BALF, and suppression of PDE1 significantly reduced inflammatory cells, including eosinophils. The effect on inflammatory cells was similar to previous reports using PDE 4 inhibitors.

The histological analysis of lung tissues also supported the hypothesis that OVA caused the infiltration of inflammatory cells, especially eosinophils, into lung tissue. Structural remodel-ing and mucus hypersecretion were also caused by OVA exposure through inflammation. Both IBMX and Vinp treatment, respectively, relieved the increased inflammatory cell infiltra-tion and structural remodeling. PAS staining and Congo red staining also showed mucus hypersecretion in bronchioles and eosinophil infiltration into the lungs. Although the absolute quantification of PAS and Congo red staining was not possible because of the difficulty of quantifying the total volume or total cell numbers, these results support the pathophysiological changes. As mentioned above, this increase in inflammatory cells was related to PDE, espe-cially PDE1, and the inhibition of PDE reduced inflammation and structural remodeling in the lungs.

In allergic inflammation, the reactions to allergens, including inflammatory cell infiltration, mucin secretion, and increased IgE levels in serum, result from various inflammatory cyto-kines secreted by activated Th2 cells [48]. In previous research, among the PDE inhibitors, PDE 4 inhibitors were reported to affect the inhibition of inflammatory cytokines of BALF (IL-4, 5 and 13) [43]. In our study, the release of IL-13 in BALF was increased significantly by OVA exposure, and the mRNA expression levels in lung tissues increased significantly. Vinp treatment significantly decreased the IL-13 release in BALF and mRNA expression in the lungs compared with the OVA group. These results suggest that IL-13 secretion and expres-sion are closely related to the PDE1 subtype. IL-4 and IL-5 were also measured, but they did not show significant differences between groups (S2 Fig). In the asthma mouse model, the changes in Th2 cytokines depend on the induction method and schedule of the model [49]. In previous work, IL-13 alone induced lung inflammation, mucus hypersecretion, and chemo-kine production [50]. In the current study, IL-13 predominated in its contribution to the pathophysiology in asthma, and the regulation of IL-13 expression and release by Vinp con-tributed to alleviating the pathophysiological changes.

In patients with asthma, the IgE level in the blood is increased systemically, which activates mast cells that release histamine and other cytokines [51]. The released histamine and

cytokines then induce allergic inflammation in the lungs [52]. B cells play a critical role in the Th2-type immune response and eosinophilic airway inflammation [53]. It is also known that IL-13 contributes to IgE switching and production from B cells [54]. In the current study, Vinp inhibited OVA-induced IL-13 expression and release. The OVA-specific IgE level in the blood increased significantly in the OVA-exposed group, and Vinp significantly decreased the increased IgE level. These results suggested that the down-regulation of IL-13 production by Vinp reduced IgE production from B cells. It can be assumed that through this process, Vinp can reduce allergic inflammation in the lungs.

MIP-1β is a well-known chemokine produced by various cells, such as neutrophils, epithelial cells, B cells, T cells, and eosinophils [55]. In a previous study, MIP-1β was found to have chemo-attractant activity for eosinophils [56]. These chemo-attractant activities of MIP-1β can cause the infiltration of eosinophils into the lungs and bronchi, which results in allergic lung inflammation [8]. In the current study, both IBMX and Vinp, respectively, inhibited the increase in MIP-1β expression level in lung tissues and its release in BALF. This result indicates that PDE1 is mainly involved in the regulation of MIP-1β expression and release. Therefore, the reduction in eosinophil infiltration into the lungs is likely caused by decreased production of MIP-1β via PDE inhibition.

The present study found similar effects between IBMX and Vinp on allergic lung inflammation, such as increased inflammatory cells, histological changes, and MIP-1β expression level. In light of the similarity between the effects of PDE1-specific inhibitor and non-specific PDE inhibitor, PDE1 is likely a substantial contributor to allergic lung inflammation.

To further support the correlation between PDE1 and allergic lung inflammation, we also demonstrated that the increase in PDE1 expression in the lungs was caused by OVA exposure. Through previous studies, it was already demonstrated that PDE4A, 4C, and 4D expression level was increased in allergic pulmonary inflammatory conditions [57, 58]. These research showed the PDE4 was associated with asthmatic conditions. In this study, PDE1A, 1B, and 1C mRNA expression levels and protein expression levels in the lungs were all increased. These results show that PDE1 expression is also associated with asthma. PDE1 is distributed in pulmonary arterial smooth muscle cells, epithelial cells, fibroblasts, macrophages, and lymphocytes [21, 25]. PDE1 inhibitors have been associated with reactive oxygen species-mediated lung inflammation via the effect on bronchial epithelial cells and macrophages, besides transforming growth factor-beta (TGF-β)-induced myofibroblastic conversion of fibroblasts in the lungs [59, 60]. Such evidence and our study suggested that PDE1 is associated with allergic lung inflammation. Thus, Vinp, a PDE1 inhibitor, can affect asthma through PDE1A, 1B, and 1C inhibition.

In conclusion, the PDE1 inhibitor alleviated asthma symptoms, such as airway hyper-responsiveness, lung inflammation, eosinophil recruitment, and mucin secretion, by reducing Th2 cytokines and MIP-1β. OVA exposure also induced PDE1A, 1B, and 1C expression. Although this study investigated the relationship between PDE1 and allergic lung inflammation, the specific mechanism remains unclear. Through further experiments, such as using PDE1 knockout mice, the relationship between PDE1 and allergic lung inflammation should be fully investigated. Despite this limitation, the hypothesis that PDE1 contributes to allergic lung inflammation and is a potential therapeutic target for asthma treatment was supported by this study.

## Supporting information

**S1 Fig. Measurement of tidal volume of lungs.** The methacholine test was performed to measure tidal volume. Mice in the OVA, OVA + IBMX, OVA + Vinp, and OVA + Dex groups

were exposed to methacholine (4, 8, 16 mg/ml). After methacholine exposure, the tidal volume of the lungs was measured by plethysmography. Data are expressed as mean ± SEM. Statistical analysis was performed using the Student's *t*-test, one-way ANOVA, and two-way ANOVA. OVA, ovalbumin; IBMX, 3-isobutyl-1-methylxanthine; Vinp, vinpocetine; Dex, dexamethasone.
(TIF)

**S2 Fig. The release of IL-4 and IL-5 in BALF.** (**A**) and (**B**) The release of IL-4 and IL-5 in BALF was measured by ELISA. Data are expressed as mean ± SEM. Statistical analysis was performed using one-way ANOVA. BALF, broncho-alveolar lavage fluid; OVA, ovalbumin; Vinp, vinpocetine; Dex, dexamethasone.
(TIF)

**S1 Raw images. Uncropped western blot data.**
(PDF)

## Author Contributions

**Conceptualization:** Won Seok Choi, Ji-Yun Lee.

**Data curation:** Won Seok Choi, Hyun Sik Kang, Hong Jo Kim, Wang Tae Lee.

**Formal analysis:** Hong Jo Kim, Wang Tae Lee.

**Investigation:** Won Seok Choi, Hyun Sik Kang, Hong Jo Kim, Wang Tae Lee.

**Methodology:** Won Seok Choi, Hong Jo Kim.

**Project administration:** Uy Dong Sohn, Ji-Yun Lee.

**Resources:** Ji-Yun Lee.

**Supervision:** Uy Dong Sohn, Ji-Yun Lee.

**Writing – original draft:** Won Seok Choi.

**Writing – review & editing:** Uy Dong Sohn, Ji-Yun Lee.

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
