## [Decision Letter · Decision Letter 0]

11 Nov 2020

PONE-D-20-32932

Vinpocetine alleviates lung inflammation via macrophage inflammatory protein-1β inhibition in an ovalbumin-induced allergic asthma model

PLOS ONE

Dear Dr. Lee,

Thank you for submitting your manuscript to PLOS ONE. After careful consideration, we feel that it has merit but does not fully meet PLOS ONE’s publication criteria as it currently stands. Therefore, we invite you to submit a revised version of the manuscript that addresses the points raised during the review process.

Two expert reviewers from the field raised a number of points that should be addressed when revising the manuscript. In particular, the role of the other Th2 cytokines should be addressed throughout the manuscript, i.e. including the Introduction and Discussion section. Moreover, since you analysed the levels of IL-4, IL-5, IL-13, MIP-1β, and interferon-gamma (IFN-γ) in BALF, all the data should be presented - at least in the supplement. Further, I ask you to expand your discussion to include additional references (see suggestions by reviewer #2) allowing you to discuss the potential molecular mechanism. It will be mandatory that you submit a detailed point-by-point response to all the comments raused by the reviewers and me.

We look forward to receiving your revised manuscript.

Kind regards,

Heinz Fehrenbach

Academic Editor

PLOS ONE

Additional Editor Comments:

In addition to the comments of our reviewers, I have some comments to the quantitative analysis of histopathological changes.

1) regarding the analysis of mucus, which was done as follows "After periodic acid–Schiff (PAS) staining of the lung tissues using the PAS Stain Kit (Abcam, Cambridge, UK), mucus secretion was evaluated by measuring the bronchoalveolar red-stained regions using ImageJ software program (NIH Image, MD, USA)": in any tissue section (2-dimensional), the area occupied by a specific tissue component depends on the total volume of this component. Consequently, you cannot distinguish between an increase in mucus due to an increase in the mucus volume stored in the cells and an increase the number of mucus producing / storing goblet cells. Please take into account this limitation of your methodological approach when discussing the results derived from this analysis.

2) regarding the analysis of eosinophils, which was done as follows "Eosinophils were counted in an area of 20,000 µm2 of lung tissues after staining with Congo red using the Congo Red Stain Kit (Abcam, Cambridge, UK).": Doing the analysis in a two-dimensional section, you in fact counted the number of cell profiles, but clearly not the cell numbers. The cell profile counts again depend on the volume of the cells analysed. Therefore, you cannot distinguish between an increase in cell numbers (due to proliferation or infiltration) and an increase in cell volume (due to hypertrophy). I therefore suggest, that you clearly indicate this limitation, or that you use a similar scoring system as you had implemented for estimating the inflammation in the tissues done in the H & E stainings.

Journal Requirements:

2. Thank you for including your ethics statement: ' All experimental procedures were in accordance with the guidelines established by the Institutional Animal Care and Use Committee of Chung-Ang University (IACUC 2019-00031).'

(a) Please amend your current ethics statement to include the full name of the ethics committee/institutional review board(s) that approved your specific study.  

(b) Once you have amended this/these statement(s) in the Methods section of the manuscript, please add the same text to the “Ethics Statement” field of the submission form (via “Edit Submission”).

For additional information about PLOS ONE ethical requirements for human subjects research, please refer to https://journals.plos.org/plosone/s/animal-research.

3. To comply with PLOS ONE submissions requirements, please provide the method of euthanasia in the Methods section of your manuscript.

4. At this time, we request that you  please report additional details in your Methods section regarding animal care, as per our editorial guidelines:

(1) Please describe the post-operative care received by the animals, including the frequency of monitoring and the specific clinical, physiological and behavioural criteria used to assess animal health and well-being.

Thank you for your attention to these requests.

5. In the Methods section, please provide the product number and any lot numbers of the primary antibodies purchased from chemical companies for your study.

6. To comply with PLOS ONE submission guidelines, in your Methods section, please provide additional information regarding your statistical analyses, including the name and version of the software used for the statistical analyses. For more information on PLOS ONE's expectations for statistical reporting, please see https://journals.plos.org/plosone/s/submission-guidelines.#loc-statistical-reporting.

7. At this time, we ask that you please provide scale bars on the microscopy images presented in Figure 1 and 2 and refer to the scale bar in the corresponding Figure legend.

8. PLOS ONE now requires that authors provide the original uncropped and unadjusted images underlying all blot or gel results reported in a submission’s figures or Supporting Information files. This policy and the journal’s other requirements for blot/gel reporting and figure preparation are described in detail at https://journals.plos.org/plosone/s/figures#loc-blot-and-gel-reporting-requirements and https://journals.plos.org/plosone/s/figures#loc-preparing-figures-from-image-files. When you submit your revised manuscript, please ensure that your figures adhere fully to these guidelines and provide the original underlying images for all blot or gel data reported in your submission. See the following link for instructions on providing the original image data: https://journals.plos.org/plosone/s/figures#loc-original-images-for-blots-and-gels.

Reviewers' comments:

Reviewer's Responses to Questions

**Comments to the Author**

1. Is the manuscript technically sound, and do the data support the conclusions?

Reviewer #1: Partly

Reviewer #2: Yes

2. Has the statistical analysis been performed appropriately and rigorously? 

Reviewer #1: Yes

Reviewer #2: Yes

3. Have the authors made all data underlying the findings in their manuscript fully available?

Reviewer #1: Yes

Reviewer #2: Yes

4. Is the manuscript presented in an intelligible fashion and written in standard English?

Reviewer #1: Yes

Reviewer #2: Yes

5. Review Comments to the Author

Reviewer #1: The manuscript number PONE-D-20-32932, entitled "Vinpocetine alleviates lung inflammation via macrophage inflammatory protein-1β inhibition in an ovalbumin-induced allergic asthma model"generally is well written and describes an interesting relationship between PDE 1 inhibition and reduction of lung inflammation in mice. Nevertheless, in my opinion manuscript requires minor corrections before further processing. Please find some comments that should be considered in the revision version of the manuscript.

Introduction:

1. In the introduction part, the Authors write that the treatment of asthma is mainly based on the use of glucocorticosteroids. I believe that this is too simplified and requires supplementing with current pharmacotherapy.

2. Also in the introduction, the authors mention the role of IL-13 as the leading one. However, the role of the remaining TH2 cytokines and their role in driving chronic asthma inflammation cannot be overlooked.

3. Third paragraph of the introduction: Please add citations confirming the role of eosinophils in inflammatory diseases. It is also worth supplementing the information on eosinophilia in asthma - focus on this disease entity.

4. Phosphodiesterases paragraph, second sentence, this is unclear. Please explain and provide the reason why PDE inhibitors are being tested as a potential therapy.

5. What kind of relationship between PDE1 and allergic lung inflammation?

Materials and methods:

1. Lack of DEX origin in materials part, how the all compounds were prepared/dissolved, what was the stock solution/concentration used in the study?

2. Does treatment with methacholine affect the parameters of inflammation? Was airway resistance measured within the same groups as the inflammatory parameters?

3. Did you try to compare resistance of the lungs also with untreated group?

4. General: Please supplement the material and method parts with the number of repeats in each experiment: eg. WB, PCR repeats, number of cells counted, fields of view, number of biological and technical repeats.

5. Please add antibodies catalog numbers used in the Immunofluorescence and Western Blot. How much protein was applied for electrophoresis. How much of RNA and cDNA was used in the qPCR experiments ?

Results and discussion

1. It would be useful to add comparisons between the VIN and IBMX groups in the description of the results. This would help to draw conclusions about the significance between these groups and conclude on superiority / or no superiority of the PDE1 inhibitor over the non-specific IBMX inhibitor. Data available in the literature show that the activity of vinpocetin and IBMX on PDE1 may be similar. The authors checked the expression of PDE1 in the lung tissue of mice with induced allergic asthma. Did the Authors also check the expression of other PDE isoforms? Are there any changes in the other isoforms? It would be interesting to see if the expression of other PDEs is also increasing in the asthmatic model. To clearly state that PDE1 is mainly responsible for the described changes, it is worth considering the implementation of knockout experiments in the future.

2. Third result: Have the other TH2 cytokines been studied? If only IL-13 has been tested then this title is too general, it is known that the response may be varied and heterogeneous.

3. Results description and MIP-1beta – the lack of DEX group description/comparison. Were the protein levels also tested?

4. The discussion needs improvement. Do not duplicate the results, but rather relate them to the current literature. Regarding the results for MIP-1beta, the differences between VIN and IBMX are poorly marked, so can we conclude that the effects depend only on PDE1 inhibition? It is also worth adding directions for further research and work limitations.

Reviewer #2: In the present manuscript Lee et. al. analyzed the therapeutic effectiveness of the PDE1 inhibitor vinpocetine in a murine model of allergic asthma.

Animals were systemically sensitized with OVA/Alum, to induce the allergic airway disease the animals were challenged by 5 times performed nebulization with OVA.

1 hr prior each challenge the animals were treated with the PDE1 inhibitor vinpocetine (Vinp), a general non specific PDE inhibitor (IBMX) or with dexamethasone (Dex).

Animals treated with Vinp demonstrated attenuated asthmatic phenotype. Asthma hallmarks like, lung function, cellular infiltrates in BAL and lung tissue and goblet cell metaplasia were improved upon treatment with Vinp in comparison to untreated animals.

Moreover, the group demonstrated reduced levels of OVA specific IgE in serum and reduced protein concentrations of IL-13 in BAL and reduced mRNA signals of the Th2 cytokine in lung tissue.

Finally the authors demonstrate that MIP-1ß was downregulated in BAL and lung tissue.

The authors conclude that the PDE1 inhibitor vinpocetine suppress the asthma phenotype by reducing Th2 cytokines and MIP-1ß.

The paper is well and comprehensibly written. It demonstrates that PDE1 proteins are upregulated in asthma and that specific blocking of PDE1 by vinpocetine is more effective as unspecific blocking of PDEs with IBMX.

Analyses in the OVA model sound solid unfortunately information upon the mode of action of vinpocetine are missing to the greatest extant.

Major:

1. The discussion exclusively focuses on the own observation of the PDE1 inhibitor vinpocetine in the murine model of allergic airway diseases without cross referencing other studies which analyzed the mode of action of vinpocetine or studies analyzing the effect of other PDE inhibitors in asthma.

This cross references will contribute to understand the possible mechanisms of the PDE1 inhibitor.

In the lung PDE1 inhibitors are associated with smooth muscle cells, oxidative stress of bronchial epithelial cells and macrophages [Brown 2007], TGFb induced fibroblast conversion [Dunkern 2007] and other mechanisms.

PDE inhibitors affect immune cells please cross reference.

2. Please mention shortly downsides of the chosen model:

a. Treatment was performed during primary challenge phase; here the lung disease is developing for the first time. It is more an acute model than a therapeutic model.

3. How do you explain the strong effect on OVA specific IgE? Are B cells affected, is it more a systemic effect? How are other immunoglobulins affected?

4. Are other immune cells affected by the inhibition of PDE1.

5. Are other side effects observable upon the systemic application of the inhibitors?

Minor:

1. Please provide information of the solvent of IBMX, Vinp, or Dex and the total volume injected.

2. Please add treatments in the schedule for inducing asthma (Fig. 1A)

3. Please specify the “airway resistance” measured with a double chamber plethysmograph.

4. Please specify how total protein concentrations for OVA specific IgE and Cytokines were calculated.

5. Please specify how relative mRNA expression was determined.

6. Please enlarge y-axis labeling of almost all graphs.

6. PLOS authors have the option to publish the peer review history of their article (what does this mean?). If published, this will include your full peer review and any attached files.

Reviewer #1: No

Reviewer #2: No

---

## [Author Response · Author response to Decision Letter 0]

15 Dec 2020

PLOS ONE

Manuscript number: PONE-D-20-32932

Title: Vinpocetine alleviates lung inflammation via macrophage inflammatory protein-1β inhibition in an ovalbumin-induced allergic asthma model

Revision date: December 12, 2020

We would like to thank you and the reviewers for the comments on our manuscript, which is entitled as “Vinpocetine alleviates lung inflammation via macrophage inflammatory protein-1β inhibition in an ovalbumin-induced allergic asthma model” We revised the manuscript according to the reviewer’s helpful comments. We made red color revised part. For the convenience, line number was added in left side in the manuscript. This revised manuscript was received English proofreading by ESSAY REVIEW, and an editorial certificate was attached. We hope that this revised manuscript can be suitable for PLoS ONE’s style requirements.

Additional Editor Comments:

In addition to the comments of our reviewers, I have some comments to the quantitative analysis of histopathological changes.

1) regarding the analysis of mucus, which was done as follows "After periodic acid–Schiff (PAS) staining of the lung tissues using the PAS Stain Kit (Abcam, Cambridge, UK), mucus secretion was evaluated by measuring the bronchoalveolar red-stained regions using ImageJ software program (NIH Image, MD, USA)": in any tissue section (2-dimensional), the area occupied by a specific tissue component depends on the total volume of this component. Consequently, you cannot distinguish between an increase in mucus due to an increase in the mucus volume stored in the cells and an increase the number of mucus producing / storing goblet cells. Please take into account this limitation of your methodological approach when discussing the results derived from this analysis.

→ Thank you for your comment. To quantify mucus production, mucus secretion was evaluated by measuring the broncho-alveolar red-stained regions using ImageJ software based on previous research. However, we agree with and discuss the limitation in the Discussion section. We also revised the relevant method to clarify the method of quantification.

Page 9, Lines 177-179

To quantify mucus production, mucus secretion was evaluated by measuring the broncho-alveolar red-stained regions using the ImageJ software program (NIH Image, Bethesda, MD, USA) [37].

Page 16, Lines 356–360

PAS staining and Congo red staining also showed mucus hypersecretion in bronchioles and eosinophil infiltration into the lungs. Although the absolute quantification of PAS and Congo red staining was not possible because of the difficulty of quantifying the total volume or total cell numbers, these results support the pathophysiological changes.

References

37. Oliveira TL, Candeia-Medeiros N, Cavalcante-Araújo PM, Melo IS, Fávaro-Pípi E, Fátima LA, et al. SGLT1 activity in lung alveolar cells of diabetic rats modulates airway surface liquid glucose concentration and bacterial proliferation. Scientific Reports. 2016;6(1):21752.http://doi.org/10.1038/srep21752 PMID: 26902517

2) regarding the analysis of eosinophils, which was done as follows "Eosinophils were counted in an area of 20,000 µm2 of lung tissues after staining with Congo red using the Congo Red Stain Kit (Abcam, Cambridge, UK).": Doing the analysis in a two-dimensional section, you in fact counted the number of cell profiles, but clearly not the cell numbers. The cell profile counts again depend on the volume of the cells analysed. Therefore, you cannot distinguish between an increase in cell numbers (due to proliferation or infiltration) and an increase in cell volume (due to hypertrophy). I therefore suggest, that you clearly indicate this limitation, or that you use a similar scoring system as you had implemented for estimating the inflammation in the tissues done in the H & E stainings.

→ Thank you for your comment. To quantify the number of eosinophils in Congo red staining, we referred to previous research and counted the eosinophils in an area of 20,000 µm2. We also agree with and discuss the limitation in the Discussion and revised the relevant Methods section to clarify the quantification procedure.

Page 9, Lines 179–182

Congo red staining of the lung tissues was performed using the Congo Red Stain Kit (Abcam). Eosinophils were counted in an area of 20,000 µm2 of lung tissues after staining with Congo red [38]. Six random fields of each stained tissue section……

Page 16, Lines 356-360

PAS staining and Congo red staining also showed mucus hypersecretion in bronchioles and eosinophil infiltration into the lungs. Although the absolute quantification of PAS and Congo red staining was not possible because of the difficulty of quantifying the total volume or total cell numbers, these results support the pathophysiological changes.

References

38. Albert EJ, Duplisea J, Dawicki W, Haidl ID, Marshall JS. Tissue eosinophilia in a mouse model of colitis is highly dependent on TLR2 and independent of mast cells. Am J Pathol. 2011;178(1):150-60.http://doi.org/10.1016/j.ajpath.2010.11.041 PMID: 21224053

Journal Requirements:

→ Thank you for your comment. Accordingly, we revised all parts of the manuscript to comply with PLoS ONE’s style requirements.

2. Thank you for including your ethics statement: ' All experimental procedures were in accordance with the guidelines established by the Institutional Animal Care and Use Committee of Chung-Ang University (IACUC 2019-00031).'

(a) Please amend your current ethics statement to include the full name of the ethics committee/institutional review board(s) that approved your specific study.

→ Thank you for your comment. We already mentioned the full name of the ethics committee that approved our study (the Institutional Animal Care and Use Committee of Chung-Ang University, Seoul, South Korea). However, we wrote the incorrect IACUC number. We amended the IACUC number. The IACUC Protocol Approval document is presented below.

Page 6 and 7, Lines 123-130

All experimental procedures were in accordance with the guidelines established by the Institutional Animal Care and Use Committee of Chung-Ang University (IACUC 2018-00124).

(b) Once you have amended this/these statement(s) in the Methods section of the manuscript, please add the same text to the “Ethics Statement” field of the submission form (via “Edit Submission”).

For additional information about PLOS ONE ethical requirements for human subjects research, please refer to https://journals.plos.org/plosone/s/animal-research.

→ Thank you for your comment. We included the ethics statement in the Methods section and added the same text to the “Ethics Statement” field of the submission form.

3. To comply with PLOS ONE submissions requirements, please provide the method of euthanasia in the Methods section of your manuscript.

→ Thank you for your helpful comment. Accordingly, we included the method of euthanasia in the Methods section, as follows.

Pages 6, Lines 126-128

After the experiment, mice were euthanized by intramuscular injection with 40 mg/kg Zoletil 50 (125 mg tiletamine and 125 mg zolazepam; Virbac, Carros, France)–10 mg/kg xylazine (Sigma Aldrich).

4. At this time, we request that you please report additional details in your Methods section regarding animal care, as per our editorial guidelines:

→ Thank you for your helpful comment. Accordingly, we included additional details regarding animal care in the Methods section.

Pages 6, Lines 119-128

Balb/c male mice (5 weeks old, weighing 20–30 g) were obtained from Samtako Bio Korea, Gyeonggi-do, Republic of Korea, and housed in the mouse facility in the R&D Center of Chung-Ang University (Seoul, Republic of Korea) with sterilized bedding. The air condition was maintained at 24±2 ℃ with 50±5% humidity, and the light/dark cycle was synchronized to 12:12 h. Pathogen-free food and water were provided. The mice were acclimated for 1 week, and 10 mice per group were randomized to five treatment groups (control, OVA, OVA + IBMX, OVA + Vinp, and OVA + Dex). Overall health was monitored twice a week. After the experiment, mice were euthanized by intramuscular injection with 40 mg/kg Zoletil 50 (125 mg tiletamine and 125 mg zolazepam; Virbac, Carros, France)–10 mg/kg xylazine (Sigma Aldrich). 

(1) Please describe the post-operative care received by the animals, including the frequency of monitoring and the specific clinical, physiological and behavioural criteria used to assess animal health and well-being.

→ Thank you for your comment. In our study, we did not perform any surgery. Instead, we monitored twice a week to check the overall health of the mice following the drug injection or OVA nebulization.

Page 6, Line 125–126

Overall health was monitored twice a week.

5. In the Methods section, please provide the product number and any lot numbers of the primary antibodies purchased from chemical companies for your study.

→ Thank you for your comment. Accordingly, we added the product number and lot numbers of the antibodies we used in this study.

Page 9, Lines 187-190

MIP-1β protein was detected using 20 µg/ml rabbit anti-CCL4 polyclonal antibody (Invitrogen, Carlsbad, CA, USA, catalog number PA5-34509, lot #UL2898185) and 24 µg/ml goat anti-rabbit IgG FITC secondary antibody (Invitrogen, catalog number 65-6111, lot #UG285467) in the 4-µm paraffin section of mouse lung tissue (n = 6).

Pages 11 and 12, Lines 244-251

The primary antibodies used were all purchased from Santa Cruz Biotechnology, Inc., and included the following: mouse anti-PDE1A antibody (catalog number sc-374602, lot #H1219, dilution ratio 1:200), mouse anti-PDE1B antibody (catalog number sc-393112, lot #E2218, dilution ratio 1:200), mouse anti-PDE1C antibody (catalog number sc-376474, lot #F2717, dilution ratio 1:200), and mouse anti-β-actin antibody (catalog number sc-47778, lot #A2317, dilution ratio 1:1000). HRP-linked horse anti-mouse IgG antibody was used as the secondary antibody (Cell Signaling Technology, Inc., Danvers, MA, USA, catalog number 7076S, lot #33, dilution ratio 1:3000 ).

6. To comply with PLOS ONE submission guidelines, in your Methods section, please provide additional information regarding your statistical analyses, including the name and version of the software used for the statistical analyses. For more information on PLOS ONE's expectations for statistical reporting, please see https://journals.plos.org/plosone/s/submission-guidelines.#loc-statistical-reporting.

→ Thank you for your comment. Accordingly, we reported the details of the software used to perform the statistical analysis in this study.

Page 12, Line 258

Data were statistically analyzed by the Student's t-test, one-way ANOVA, and two-way ANOVA using GraphPad Prism 7 (GraphPad Software, Inc., San Diego, CA, USA).

7. At this time, we ask that you please provide scale bars on the microscopy images presented in Figure 1 and 2 and refer to the scale bar in the corresponding Figure legend.

→ Thank you for your comment. Accordingly, we revised all microscopy images to include scale bars. We also mentioned the scale bar in the legends of Figs 1E (below) and 2A�2C (overleaf).

Page 29, Lines 627–629 (Fig 1, legend)

(E) Stained cells were observed using a microscope) (red arrows, eosinophils; black arrows, macrophages; magnification, 63×; scale bar, 10 µm).

Pages 29, Lines 638-641 (Fig 2, legend)

H&E and PAS staining (left-hand side: magnification 10×; scale bar, 100 μm; right-hand side: magnification 40×; scale bar, 30 μm)…… Congo red staining (magnification 63×, scale bar, 20 μm)……

In Fig 4C, the white line in each image indicates the scale bar.

Page 30, Lines 662 and 663

(C) MIP-1β expression in lung tissue was detected by immunofluorescence staining (magnification, 63×; scale bar, 20 μm).

8. PLOS ONE now requires that authors provide the original uncropped and unadjusted images underlying all blot or gel results reported in a submission’s figures or Supporting Information files. This policy and the journal’s other requirements for blot/gel reporting and figure preparation are described in detail at https://journals.plos.org/plosone/s/figures#loc-blot-and-gel-reporting-requirements and https://journals.plos.org/plosone/s/figures#loc-preparing-figures-from-image-files. When you submit your revised manuscript, please ensure that your figures adhere fully to these guidelines and provide the original underlying images for all blot or gel data reported in your submission. See the following link for instructions on providing the original image data: https://journals.plos.org/plosone/s/figures#loc-original-images-for-blots-and-gels.

→ Thank you for your comment. We already submitted the original uncropped and unadjusted images of the blot data. We attached the images and provided the data as Supporting information files, as follows:

  

Reviewers' comments:

Reviewer #1: The manuscript number PONE-D-20-32932, entitled "Vinpocetine alleviates lung inflammation via macrophage inflammatory protein-1β inhibition in an ovalbumin-induced allergic asthma model"generally is well written and describes an interesting relationship between PDE 1 inhibition and reduction of lung inflammation in mice. Nevertheless, in my opinion manuscript requires minor corrections before further processing. Please find some comments that should be considered in the revision version of the manuscript.

Introduction:

1. In the introduction part, the Authors write that the treatment of asthma is mainly based on the use of glucocorticosteroids. I believe that this is too simplified and requires supplementing with current pharmacotherapy.

→ Thank you for your helpful comment. Accordingly, the current pharmacotherapy of asthma was reported in the Introduction section

Page 3, Lines 39-47

Treatment of asthma is usually dependent on corticosteroids and beta2-agonist as a bronchodilator [3]. Some biologic agents, including anti-IgE and anti-interleukin-5 (IL-5), have been developed recently and used in the treatment of severe asthma [4]. However, the specific mechanisms of asthma remain unclear. Although inhaled corticosteroids are the gold-standard therapy used to treat patients with asthma, long-term use of high-dose inhaled corticosteroids can cause adverse effects, such as hypothalamic–pituitary–adrenal axis suppression, reduced bone growth, and increased risk of opportunistic infections [5]. Therefore, there is still a need for the development of new asthma treatment.

References

3. Hogan AD, Bernstein JA. GINA updated 2019: Landmark changes recommended for asthma management. Ann Allergy Asthma Immunol. 2020;124(4):311-3.https://doi.org/10.1016/j.anai.2019.11.005 PMID: 31734328

4. Lommatzsch M, Buhl R, Korn S. The treatment of mild and moderate asthma in adults. Dtsch Arztebl Int. 2020;117(25):434-44.http://doi.org/10.3238/arztebl.2020.0434 PMID: 32885783

5. Qian J, Ma X, Xun Y, Pan L. Protective effect of forsythiaside A on OVA-induced asthma in mice. Eur J Pharmacol. 2017;812:250-5.http://doi.org/10.1016/j.ejphar.2017.07.033 PMID: 28733217

2. Also in the introduction, the authors mention the role of IL-13 as the leading one. However, the role of the remaining TH2 cytokines and their role in driving chronic asthma inflammation cannot be overlooked.

→ Thank you for your helpful comment. Accordingly, besides IL-13, we reported the role of other Th2 cytokines, such as IL-4 and IL-5, in driving chronic asthma inflammation in the Introduction.

Page 3, Lines 48-54

Asthma has been associated with T helper cell 2 (Th2)-mediated immunity due to aberrant production of IL-4, IL5, and IL13. In one study, 50% of asthmatic patients showed Th2-related inflammation [6]. Atopic asthma and the genetic predisposition to produce immunoglobulin E (IgE) to common allergens is driven by IL-4-dependent Ig class switching in B cells [7]. Airway eosinophilia depends on both IL-5 and Stat6 signaling [8]. Each cytokine has distinct functional effects in the induction of disease, but IL-13 predominates in its contribution to the pathophysiology in asthma [9].

References

7. Lambrecht BN, Hammad H, Fahy JV. The cytokines of asthma. Immunity. 2019;50(4):975-91.http://doi.org/10.1016/j.immuni.2019.03.018 PMID: 30995510

8. Mattes J, Yang M, Mahalingam S, Kuehr J, Webb DC, Simson L, et al. Intrinsic defect in T cell production of interleukin (IL)-13 in the absence of both IL-5 and eotaxin precludes the development of eosinophilia and airways hyperreactivity in experimental asthma. J Exp Med. 2002;195(11):1433-44.http://doi.org/10.1084/jem.20020009 PMID: 12045241

9. Elias JA, Lee CG, Zheng T, Ma B, Homer RJ, Zhu Z. New insights into the pathogenesis of asthma. J Clin Invest. 2003;111(3):291-7.http://doi.org/10.1172/jci17748 PMID: 12569150

3. Third paragraph of the introduction: Please add citations confirming the role of eosinophils in inflammatory diseases. It is also worth supplementing the information on eosinophilia in asthma - focus on this disease entity.

→ Thank you for your helpful comment. Accordingly, we added citations to confirm the role of eosinophils and supplement the information on eosinophilia in asthma.

Pages 3 and 4, Lines 58-60

In most asthma phenotypes, there are increases in eosinophils in the tissues, blood, and bone marrow and in general, the numbers increase with disease severity [12]. The eosinophil is the central effector cell responsible for ongoing airway inflammation [12, 13].

References

12. Kay AB. The role of eosinophils in the pathogenesis of asthma. Trends Mol Med. 2005;11(4):148-52.https://doi.org/10.1016/j.molmed.2005.02.002 PMID: 15823751

13. Kim S-H, Kim B-K, Lee Y-C. Effects of Corni fructus on ovalbumin-induced airway inflammation and airway hyper-responsiveness in a mouse model of allergic asthma. J Inflamm. 2012;9(1):9.http://doi.org/10.1186/1476-9255-9-9 PMID: 22439901

4. Phosphodiesterases paragraph, second sentence, this is unclear. Please explain and provide the reason why PDE inhibitors are being tested as a potential therapy.

→ Thank you for your helpful comment. To clarify why PDE inhibitors are being tested as a potential therapy, we revised the PDE paragraph in the Introduction. In brief, PDE inhibitors prevent the inactivation of intracellular cAMP and cGMP. Studies have demonstrated that cAMP and cGMP have a role in airway smooth muscle relaxation and down-regulate the airway inflammation and airway remodeling. In addition, most PDEs are expressed in lung and immune cells. In this context, we hypothesized that targeting PDEs has potential application in asthma treatment.

Page 4, Lines 71–82

Phosphodiesterase (PDE) inhibitors prevent the inactivation of intracellular cyclic adenosine monophosphate (cAMP) and cyclic guanosine monophosphate (cGMP) [19]. Studies have demonstrated that cAMP and cGMP have a role in airway smooth muscle relaxation and down-regulate the airway inflammation and airway remodeling [20, 21]. Consequently, most PDEs are expressed in lung and immune cells [21]. The cAMP-specific PDE family negatively regulates the function of almost all pro-inflammatory and immune cells and exerts widespread anti-inflammatory activity in animal models of asthma. Some PDE inhibitors have been implicated in the treatment of chronic obstructive pulmonary disease and asthma [22]. For example, Roflumilast, a PDE4 inhibitor, is currently being used to treat chronic obstructive pulmonary disease and effectively improves asthma symptoms [10, 23]. Based on this evidence, we hypothesized that targeting PDEs has potential application in asthma treatment.

References

20. Chen Y-f, Huang G, Wang Y-m, Cheng M, Zhu F-f, Zhong J-n, et al. Exchange protein directly activated by cAMP (Epac) protects against airway inflammation and airway remodeling in asthmatic mice. Respir Res. 2019;20(1):285.http://doi.org/10.1186/s12931-019-1260-2 PMID: 31852500

21. Zuo H, Cattani-Cavalieri I, Musheshe N, Nikolaev VO, Schmidt M. Phosphodiesterases as therapeutic targets for respiratory diseases. Pharmacol Ther. 2019;197:225-42.https://doi.org/10.1016/j.pharmthera.2019.02.002 PMID: 30759374

22. Page CP. Phosphodiesterase inhibitors for the treatment of asthma and chronic obstructive pulmonary disease. Int Arch Allergy Immunol. 2014;165(3):152-64.http://doi.org/10.1159/000368800 PMID: 25532037

5. What kind of relationship between PDE1 and allergic lung inflammation?

→ Thank you for your helpful comment. PDE1 is expressed in lung and immune cells, such as pulmonary arterial smooth muscle cells, airway smooth muscles, epithelial cells, fibroblasts, macrophages, and T cells. PDE1 inhibition can regulate cAMP and cGMP degradation, associated with smooth muscle contraction, airway inflammation, and airway remodeling. Furthermore, PDE1 inhibitors activate T cells and suppress IL-13 production, implicated in allergic lung inflammation. Based on this knowledge, we hypothesize that PDE1 inhibition can regulate allergic lung inflammation. To clarify the relationship between PDE1 and allergic lung inflammation, we revised the Introduction section as follows:

Page 5, Lines 83-96

PDE1, as one of the subtypes in the PDE superfamily, is expressed in pulmonary arterial smooth muscle cells, epithelial cells, fibroblasts, macrophages, and lymphocytes [21, 24, 25]. It is well-known that PDE1 degrades both cAMP and cGMP [26]. Inhibition of cAMP and cGMP degradation can regulate airway inflammation and airway smooth muscle contraction [20]. Although the direct association between PDE1 and asthma remains unclear, recent studies have provided clues to the relationship between allergic lung inflammation and PDE1. PDE1A and PDE1C protein expression was detected in the isolated lung cells of mice, and it was increased in the inflammation state [27, 28]. A previous study also reported that PDE1A inhibition prevented lung fibrosis [29]. Other work demonstrated that PDE1 inhibition may dampen inflammatory responses of microglia in the disease state [26]. Moreover, PDE1B has been associated with the activation or differentiation of immune cells [30, 31]. It was found to be expressed in T lymphocytes and modulate the allergic response by regulating IL-13 production, which was closely related to allergic lung inflammation [30-32]. In this context, we hypothesized that PDE1 inhibition could down-regulate allergic inflammation.

Materials and methods:

1. Lack of DEX origin in materials part, how the all compounds were prepared/dissolved, what was the stock solution/concentration used in the study?

→ Thank you for your helpful comment. Accordingly, we reported the DEX origin in the Materials section. We also detailed the methods used to prepare all compounds in the Materials and Methods. We prepared stock solutions of 3.5 mg/ml DEX, IBMX, and Vinp in DMSO. Before the intraperitoneal injection, DEX, IBMX, and Vinp were diluted to 1 mg/ml with normal saline.

Page 6, Lines 110–115

OVA, grade V from hen egg white (lyophilized powder, �98%) and dexamethasone (Dex; catalog number D1756) were purchased from Sigma–Aldrich (St Louis, MO, USA). Aluminum hydroxide (Imject® Alum) was purchased from Thermo Scientific, Rockford, IL, USA. IBMX and Vinp were obtained from Tocris Bioscience, Bristol, UK. Stock solutions of IBMX, Vinp, and Dex were prepared at 3.5 mg/ml in DMSO. Before the intraperitoneal (i.p.) injection to the mice, IBMX, Vinp, and Dex were diluted to 1 mg/ml with normal saline.

2. Does treatment with methacholine affect the parameters of inflammation? Was airway resistance measured within the same groups as the inflammatory parameters?

→ Thank you for your helpful comment. We performed the methacholine test to measure airway resistance. After measuring airway resistance, mice were sacrificed to obtain blood, BALF, and lung tissues. Using these samples, we measured the inflammatory parameters, such as OVA-specific IgE, IL-13, and MIP-1β. All groups, including the control group, underwent the methacholine test; therefore, we believed that comparing the inflammatory parameters between groups was possible. We have also used this protocol in previous experiments and confirmed that the methacholine test did not affect allergic lung inflammation.

Page 7, Lines 146 and 147

We also confirmed that the methacholine test did not affect the inflammatory parameter in this study.

3. Did you try to compare resistance of the lungs also with untreated group?

→ Thank you for your comment. We measured airway resistance in the control group as an untreated group. In this group, the methacholine test did not interfere with further analysis (e.g., inflammatory parameters).

→ In the present study, we did not measure the direct resistance of lung tissue. Instead, we measured tidal volume. Tidal volume did not show significant differences between groups. We added the tidal volume data as Supplementary data and discussed this result.

Page 16, Lines 340-342

In tidal volume, there were no significant differences among the groups (S1 Fig). The mouse model of asthma induced in this study does not appear to be associated with a decrease in tidal volume.

Page 33, Lines 1131-1137

S1 Fig. Measurement of tidal volume of lungs. The methacholine test was performed to measure tidal volume. Mice in the OVA, OVA + IBMX, OVA + Vinp, and OVA + Dex groups were exposed to methacholine (4, 8, 16 mg/ml). After methacholine exposure, the tidal volume of the lungs was measured by plethysmography. Data are expressed as mean ± SEM. Statistical analysis was performed using the Student's t-test, one-way ANOVA, and two-way ANOVA. OVA, ovalbumin; IBMX, 3-isobutyl-1-methylxanthine; Vinp, vinpocetine; Dex, dexamethasone.

4. General: Please supplement the material and method parts with the number of repeats in each experiment: eg. WB, PCR repeats, number of cells counted, fields of view, number of biological and technical repeats.

→ Thank you for your helpful comment. We reported the number of repeats in each experiment of the Methods section.

Page 8, Line 171 and 172

The left lung was removed from each mouse, fixed with 10% formalin solution, embedded using Tissue-Tek® (Sakura Finetek®, Torrance, CA, USA), then sectioned using a Leica microtome 820 (Leica Microsystems, Wetzlar, Germany) at 4 µm (n = 6). The sectioned tissues were stained with hematoxylin–eosin (H&E).

Page 9, Line 182

Six random fields of each stained tissue section were observed under a microscope (Leica Microsystems), and images were captured using a Leica DM 480 camera (Leica Microsystems).

Page 9, Lines 190-192

…… in the 4-µm paraffin section of mouse lung tissue (n=6). The nucleus was stained using UltraCruz® Aqueous Mounting Medium with DAPI (Santa Cruz Biotechnology, Inc., Dallas, TX, USA). Six random field of each ……

Page 9, Line 198

Blood samples were obtained from the inferior vena cava for biochemical assay (n = 10). Obtained blood samples were centrifuged at 1,500g for 10 min, and the serum was separated.

Page 10, Line 208 and 209

The levels of IL-4, IL-5, IL-13, and MIP-1β in BALF were measured using an ELISA kit (R&D Systems, Inc., Minneapolis, MN, USA) according to the manufacturer's instructions (n = 6�8 per group). Absorbance was measured as described above.

Page 11, Lines 228 and 229

The RT-qPCR analysis was performed in three independent experiments (n = 6�8 per group)

Page 12, Lines 252 and 253

Western blot analysis was performed in three independent experiments (n = 4�5 per group).

5. Please add antibodies catalog numbers used in the Immunofluorescence and Western Blot. How much protein was applied for electrophoresis. How much of RNA and cDNA was used in the qPCR experiments?

→ Thank you for your helpful comment. Accordingly, we added the catalog numbers of the antibodies used in the immunofluorescence and western blot. We also revised the Methods section to clarify the western blot and qPCR procedures.

Page 9, Lines 187–190

MIP-1β protein was detected using 20 µg/ml rabbit anti-CCL4 polyclonal antibody (Invitrogen, Carlsbad, CA, USA, catalog number PA5-34509, lot #UL2898185) and 24 µg/ml goat anti-rabbit IgG FITC secondary antibody (Invitrogen, catalog number 65-6111, lot #UG285467) in the 4-µm paraffin section of mouse lung tissue (n = 6).

Pages 11 and 12, Lines 232-251

Protein expression in the lungs was measured by western blot analysis. Proteins were extracted using RIPA buffer (Thermo Scientific) containing protease inhibitor cocktail (Roche, Basel, Switzerland) and phosphatase inhibitor cocktail (Roche), as per the manufacturer's instructions. Extracted protein concentrations were quantified by the BCA protein assay reagent (Thermo Scientific) using bovine serum albumin (BSA) as a standard. For electrophoresis, 20 μg of protein was loaded into each well. The protein was separated on a 10% sodium dodecyl sulfate-polyacrylamide gel electrophoresis (SDS-PAGE) gel at 60 V for 30 min, followed by 90 V for 2 h, and then transferred to polyvinylidene fluoride (PVDF) membranes (Merck, Darmstadt, Germany). The membranes were blocked by incubation with 5% BSA in PBS-T buffer (0.1% Tween in PBS, pH 7.6) at room temperature for 1 h to prevent non-specific binding. The membranes were incubated overnight with primary antibodies at 4 °C. Then, the membranes were incubated with secondary antibodies at room temperature for 2 h. The primary antibodies used were all purchased from Santa Cruz Biotechnology, Inc., and included the following: mouse anti-PDE1A antibody (catalog number sc-374602, lot #H1219, dilution ratio 1:200), mouse anti-PDE1B antibody (catalog number sc-393112, lot #E2218, dilution ratio 1:200), mouse anti-PDE1C antibody (catalog number sc-376474, lot #F2717, dilution ratio 1:200), and mouse anti-β-actin antibody (catalog number sc-47778, lot #A2317, dilution ratio 1:1000). HRP-linked horse anti-mouse IgG antibody was used as the secondary antibody (Cell Signaling Technology, Inc., Danvers, MA, USA, catalog number 7076S, lot #33, dilution ratio 1:3000 ).

Pages 10 and 11, Lines 214-228

The RNA was spectrophotometrically quantified using a NanoDrop ND-1000 (Thermo Fisher Scientific). One microgram of total RNA was used to synthesize cDNA using the iScript™ cDNA Synthesis Kit (Bio-Rad, Hercules, CA, USA). After synthesis, 100 ng cDNA was used for quantitative PCR (qPCR). qPCR was performed using the iQ™ SYBR® Green Supermix (Bio-Rad). The primer sequences used in this study are shown in Table 1. The CFX96 Real-Time PCR Detection System (Bio-Rad) was used to monitor fluorescent intensity during amplification. cDNA was initially denatured at 95 °C for 3 min, then cycled 40 times through the following cycle: denaturation (95 °C for 10 s), annealing (55 °C for 30 s), plate read. A melting curve was generated after cycling by increasing the temperature from 55 to 95 °C at 0.5 °C for 5 s and performing a plate read at each increment. CFX Manager software was used to automatically calculate the cycle quantification value at which samples amplified at a high enough value to be detected (ΔCt values). Each ΔCt value was normalized against GAPDH, used as a housekeeping gene. Transcript expression was determined relative to the control group.

Results and discussion

1. It would be useful to add comparisons between the VIN and IBMX groups in the description of the results. This would help to draw conclusions about the significance between these groups and conclude on superiority / or no superiority of the PDE1 inhibitor over the non-specific IBMX inhibitor. Data available in the literature show that the activity of vinpocetin and IBMX on PDE1 may be similar. The authors checked the expression of PDE1 in the lung tissue of mice with induced allergic asthma. Did the Authors also check the expression of other PDE isoforms? Are there any changes in the other isoforms? It would be interesting to see if the expression of other PDEs is also increasing in the asthmatic model. To clearly state that PDE1 is mainly responsible for the described changes, it is worth considering the implementation of knockout experiments in the future.

→ Thank you for your helpful comment. Accordingly, we compared the Vinp and IBMX groups and reviewed this result in the Discussion section. Interestingly, the effects on allergic lung inflammation were similar between Vinp and IBMX. IBMX is a non-specific PDE inhibitor, and Vinp is a PDE1-specific inhibitor. The fact that the PDE1 inhibitor had a similar level of effect to the non-specific PDE inhibitor suggests that the effect of the PDE inhibitor on allergic lung inflammation is due to PDE1. Therefore, we considered that PDE1 contributes to allergic lung inflammation and is a potential therapeutic target in asthma treatment. We also mentioned further research using PDE1 knockout mice.

Pages 18, Lines 396-404

The present study found similar effects between IBMX and Vinp on allergic lung inflammation, such as increased inflammatory cells, histological changes, and MIP-1β expression level. In light of the similarity between the effects of PDE1-specific inhibitor and non-specific PDE inhibitor, PDE1 is likely a substantial contributor to allergic lung inflammation.

To further support the correlation between PDE1 and allergic lung inflammation, we also demonstrated that the increase in PDE1 expression in the lungs was caused by OVA exposure. PDE1A, 1B, and 1C mRNA expression levels and protein expression levels in the lungs were all increased.

Page 19, Lines 415–418

Although this study investigated the relationship between PDE1 and allergic lung inflammation, the specific mechanism remains unclear. Through further experiments, such as using PDE1 knockout mice, the relationship between PDE1 and allergic lung inflammation should be fully investigated.

2. Third result: Have the other TH2 cytokines been studied? If only IL-13 has been tested then this title is too general, it is known that the response may be varied and heterogeneous.

→ Thank you for your helpful comment. We also studied other Th2 cytokines, such as IL-4 and IL-5, that are also known to contribute to allergic lung inflammation. In this study, we measured the levels of IL-4 and IL-5 released in BALF. However, we did not find any significant differences between the groups. We added the IL-4 and IL-5 data as Supplementary data. We also revised the third subtitle of the Results and Discussion sections.

Page 14, Lines 294 and 295

Down-regulation of allergic inflammation by IBMX and Vinp through OVA-induced decrease in IgE and IL-13

Page 17, Lines 370–376

IL-4 and IL-5 were also measured, but they did not show significant differences between groups (S2 Fig). In the asthma mouse model, the changes in Th2 cytokines depend on the induction method and schedule of the model [45]. In previous work, IL-13 alone induced lung inflammation, mucus hypersecretion, and chemokine production [46]. In the current study, IL-13 predominated in its contribution to the pathophysiology in asthma, and the regulation of IL-13 expression and release by Vinp contributed to alleviating the pathophysiological changes.

Page 33, Lines 1139–1142

S2 Fig. The release of IL-4 and IL-5 in BALF. (A) and (B) The release of IL-4 and IL-5 in BALF was measured by ELISA. Data are expressed as mean ± SEM. Statistical analysis was performed using one-way ANOVA. BALF, broncho-alveolar lavage fluid; OVA, ovalbumin; Vinp, vinpocetine; Dex, dexamethasone.

References

45. Epstein MM, Tilp C, Erb KJ. The use of mouse asthma models to successfully discover and develop novel drugs. Int Arch Allergy Immunol. 2017;173(2):61-70.http://doi.org/10.1159/000473699 PMID: 28586774

46. Zhu Z, Homer RJ, Wang Z, Chen Q, Geba GP, Wang J, et al. Pulmonary expression of interleukin-13 causes inflammation, mucus hypersecretion, subepithelial fibrosis, physiologic abnormalities, and eotaxin production. J Clin Invest. 1999;103(6):779-88.http://doi.org/10.1172/jci5909 PMID: 10079098

3. Results description and MIP-1beta – the lack of DEX group description/comparison. Were the protein levels also tested?

→ Thank you for your comment. We included the results of MIP-1β release in BALF and mRNA expression for the Dex group. Our data indicated that Dex treatment decreased the protein level (release in BALF). However, the mRNA expression level of MIP-1β was not significantly different between groups OVA and Dex. We thought that the mRNA expression level might differ from the protein level during post-transcriptional regulation, translational regulation, and regulation of protein degradation. Although we do not know the exact reason for the dissimilarity between the mRNA expression level and protein level in this study, we will investigate this in more detail through further study.

Page 14 and 15, Lines 316–318

Dex significantly reduced the MIP-1β release in BALF (P<0.01), but the MIP-1β mRNA expression level did not reduce significantly compared with the OVA group (Fig 4A and 4B).

4. The discussion needs improvement. Do not duplicate the results, but rather relate them to the current literature. Regarding the results for MIP-1beta, the differences between VIN and IBMX are poorly marked, so can we conclude that the effects depend only on PDE1 inhibition? It is also worth adding directions for further research and work limitations.

→ Thank you for your helpful comment. We reviewed the entire Discussion section for improvement. Especially, we explained the similarity between the effects of Vinp and IBMX on allergic lung inflammation, particularly regarding MIP-1β. We also described the study limitations in the Conclusion section.

Pages 18 and 19, Lines 396-401

The present study found similar effects between IBMX and Vinp on allergic lung inflammation, such as increased inflammatory cells, histological changes, and MIP-1β expression level. In light of the similarity between the effects of PDE1-specific inhibitor and non-specific PDE inhibitor, PDE1 is likely a substantial contributor to allergic lung inflammation.

To further support the correlation between PDE1 and allergic lung inflammation, ……

Page 19, Lines 414-420

OVA exposure also induced PDE1A, 1B, and 1C expression. Although this study investigated the relationship between PDE1 and allergic lung inflammation, the specific mechanism remains unclear. Through further experiments, such as using PDE1 knockout mice, the relationship between PDE1 and allergic lung inflammation should be fully investigated. Despite this limitation, the hypothesis that PDE1 contributes to allergic lung inflammation and is a potential therapeutic target for asthma treatment was supported by this study.

Reviewer #2: In the present manuscript Lee et. al. analyzed the therapeutic effectiveness of the PDE1 inhibitor vinpocetine in a murine model of allergic asthma.

Animals were systemically sensitized with OVA/Alum, to induce the allergic airway disease the animals were challenged by 5 times performed nebulization with OVA.

1 hr prior each challenge the animals were treated with the PDE1 inhibitor vinpocetine (Vinp), a general non specific PDE inhibitor (IBMX) or with dexamethasone (Dex).

Animals treated with Vinp demonstrated attenuated asthmatic phenotype. Asthma hallmarks like, lung function, cellular infiltrates in BAL and lung tissue and goblet cell metaplasia were improved upon treatment with Vinp in comparison to untreated animals.

Moreover, the group demonstrated reduced levels of OVA specific IgE in serum and reduced protein concentrations of IL-13 in BAL and reduced mRNA signals of the Th2 cytokine in lung tissue.

Finally the authors demonstrate that MIP-1ß was downregulated in BAL and lung tissue.

The authors conclude that the PDE1 inhibitor vinpocetine suppress the asthma phenotype by reducing Th2 cytokines and MIP-1ß.

The paper is well and comprehensibly written. It demonstrates that PDE1 proteins are upregulated in asthma and that specific blocking of PDE1 by vinpocetine is more effective as unspecific blocking of PDEs with IBMX.

Analyses in the OVA model sound solid unfortunately information upon the mode of action of vinpocetine are missing to the greatest extant.

Major:

1. The discussion exclusively focuses on the own observation of the PDE1 inhibitor vinpocetine in the murine model of allergic airway diseases without cross referencing other studies which analyzed the mode of action of vinpocetine or studies analyzing the effect of other PDE inhibitors in asthma.

This cross references will contribute to understand the possible mechanisms of the PDE1 inhibitor.

In the lung PDE1 inhibitors are associated with smooth muscle cells, oxidative stress of bronchial epithelial cells and macrophages [Brown 2007], TGFb induced fibroblast conversion [Dunkern 2007] and other mechanisms.

PDE inhibitors affect immune cells please cross reference.

→ Thank you for your helpful comment. Accordingly, we revised the Introduction and Discussion sections to explain the mode of action of PDE1 inhibitors in asthma, with cross-referencing.

Page 5, Lines 83-87

PDE1, as one of the subtypes in the PDE superfamily, is expressed in pulmonary arterial smooth muscle cells, epithelial cells, fibroblasts, macrophages, and lymphocytes [21, 24, 25]. It is well-known that PDE1 degrades both cAMP and cGMP [26]. Inhibition of cAMP and cGMP degradation can regulate airway inflammation and airway smooth muscle contraction [20].

Page 18 and 19, Lines 404–410

These results show that PDE1 expression is associated with asthma. PDE1 is distributed in pulmonary arterial smooth muscle cells, epithelial cells, fibroblasts, macrophages, and lymphocytes [21, 25]. PDE1 inhibitors have been associated with reactive oxygen species-mediated lung inflammation via the effect on bronchial epithelial cells and macrophages, besides transforming growth factor-beta (TGF-β)-induced myofibroblastic conversion of fibroblasts in the lungs [53, 54]. Such evidence and our study suggested that PDE1 is associated with allergic lung inflammation.

References

20. Chen Y-f, Huang G, Wang Y-m, Cheng M, Zhu F-f, Zhong J-n, et al. Exchange protein directly activated by cAMP (Epac) protects against airway inflammation and airway remodeling in asthmatic mice. Respir Res. 2019;20(1):285.http://doi.org/10.1186/s12931-019-1260-2 PMID: 31852500

21. Zuo H, Cattani-Cavalieri I, Musheshe N, Nikolaev VO, Schmidt M. Phosphodiesterases as therapeutic targets for respiratory diseases. Pharmacol Ther. 2019;197:225-42.https://doi.org/10.1016/j.pharmthera.2019.02.002 PMID: 30759374

24. Giembycz MA. Phosphodiesterase 4 inhibitors and the treatment of asthma: where are we now and where do we go from here? Drugs. 2000;59(2):193-212.http://doi.org/10.2165/00003495-200059020-00004 PMID: 10730545

25. Francis SH, Corbin JD. Cyclic GMP phosphodiesterases. In: Lennarz WJ, Lane MD, editors. Encyclopedia of Biological Chemistry (Second Edition). Waltham: Academic Press; 2013. p. 567-73.

26. O'Brien JJ, O'Callaghan JP, Miller DB, Chalgeri S, Wennogle LP, Davis RE, et al. Inhibition of calcium-calmodulin-dependent phosphodiesterase (PDE1) suppresses inflammatory responses. Mol Cell Neurosci. 2020;102:103449.http://doi.org/10.1016/j.mcn.2019.103449 PMID: 31770590

53. Brown DM, Hutchison L, Donaldson K, MacKenzie SJ, Dick CA, Stone V. The effect of oxidative stress on macrophages and lung epithelial cells: the role of phosphodiesterases 1 and 4. Toxicol Lett. 2007;168(1):1-6.http://doi.org/10.1016/j.toxlet.2006.10.016 PMID: 17129690

54. Dunkern TR, Feurstein D, Rossi GA, Sabatini F, Hatzelmann A. Inhibition of TGF-beta induced lung fibroblast to myofibroblast conversion by phosphodiesterase inhibiting drugs and activators of soluble guanylyl cyclase. Eur J Pharmacol. 2007;572(1):12-22.http://doi.org/10.1016/j.ejphar.2007.06.036 PMID: 17659276

2. Please mention shortly downsides of the chosen model:

a. Treatment was performed during primary challenge phase; here the lung disease is developing for the first time. It is more an acute model than a therapeutic model.

→ Thank you for your helpful comment. The schedule for sensitization and challenge in a murine OVA-induced asthma mouse model is very important because its affects the inflammation pattern. Through the pilot study, we decided to time-schedule the sensitization and challenge.

→ Furthermore, we also focused on the contribution of PDE1 in asthma and its potential as a therapeutic target. In previous research, treatment began on or before the 1st challenge to investigate the relationship between the target and asthma [5, 10, 37�39]. Based on our findings, we hypothesize that PDE1 contributes to allergic lung inflammation and is a potential therapeutic target for asthma treatment. To explain why this model was chosen, we revised the Discussion.

Page 15, Lines 332–335

This study also focused on the contribution of PDE1 to asthma and its potential as a therapeutic target. To investigate the relationship with asthma, the appropriate asthma mice model and treatment schedule were determined from previous research and a pilot study [5, 10, 39-41].

References

5. Qian J, Ma X, Xun Y, Pan L. Protective effect of forsythiaside A on OVA-induced asthma in mice. Eur J Pharmacol. 2017;812:250-5.http://doi.org/10.1016/j.ejphar.2017.07.033 PMID: 28733217

10. Park HJ, Lee JH, Park YH, Han H, Sim da W, Park KH, et al. Roflumilast ameliorates airway hyperresponsiveness caused by diet-induced obesity in a murine model. Am J Respir Cell Mol Biol. 2016;55(1):82-91.http://doi.org/10.1165/rcmb.2015-0345OC PMID: 26756251

39. Højen JF, Kristensen MLV, McKee AS, Wade MT, Azam T, Lunding LP, et al. IL-1R3 blockade broadly attenuates the functions of six members of the IL-1 family, revealing their contribution to models of disease. Nat Immunol. 2019;20(9):1138-49.http://doi.org/10.1038/s41590-019-0467-1 PMID: 31427775

40. Bao A, Li F, Zhang M, Chen Y, Zhang P, Zhou X. Impact of ozone exposure on the response to glucocorticoid in a mouse model of asthma: involvements of p38 MAPK and MKP-1. Respir Res. 2014;15:126.http://doi.org/10.1186/s12931-014-0126-x PMID: 25287866

41. Ye P, Yang XL, Chen X, Shi C. Hyperoside attenuates OVA-induced allergic airway inflammation by activating Nrf2. Int Immunopharmacol. 2017;44:168-73.http://doi.org/10.1016/j.intimp.2017.01.003 PMID: 28107754

3. How do you explain the strong effect on OVA specific IgE? Are B cells affected, is it more a systemic effect? How are other immunoglobulins affected?

→ Thank you for your helpful comment. B cells play a critical role in Th2-type immune responses and eosinophilic airway inflammation. It is also known that IL-13 contributes to IgE switching and production. In our study, Vinp inhibited OVA-induced IL-13 expression and release. Therefore, we thought that the regulation of IL-13 production reduced IgE production. We revised the Discussion to explain the relationship between IL-13 and IgE.

Page 17, Lines 379–386

B cells play a critical role in the Th2-type immune response and eosinophilic airway inflammation [49]. It is also known that IL-13 contributes to IgE switching and production from B cells [50]. In the current study, Vinp inhibited OVA-induced IL-13 expression and release. The OVA-specific IgE level in the blood increased significantly in the OVA-exposed group, and Vinp significantly decreased the increased IgE level. These results suggested that the down-regulation of IL-13 production by Vinp reduced IgE production from B cells. It can be assumed that through this process, Vinp can reduce allergic inflammation in the lungs.

In this study, we did not examine the other immunoglobulins. We will investigate the effect of Vinp on the stimulation of other immunoglobulins in the OVA-induced asthma mice model through further study.

References

49. Drake LY, Iijima K, Hara K, Kobayashi T, Kephart GM, Kita H. B cells play key roles in th2-type airway immune responses in mice exposed to natural airborne allergens. PLoS One. 2015;10(3):e0121660.http://doi.org/10.1371/journal.pone.0121660 PMID: 25803300

50. Stone KD, Prussin C, Metcalfe DD. IgE, mast cells, basophils, and eosinophils. J Allergy Clin Immunol. 2010;125(2 Suppl 2):S73-80.http://doi.org/10.1016/j.jaci.2009.11.017 PMID: 20176269

4. Are other immune cells affected by the inhibition of PDE1.

→ Thank you for your helpful comment. This study showed that macrophages and eosinophils were decreased by PDE1 inhibition. It is known that cytokines, such as IL-13, are also affected by the inhibition of PDE1, suggesting that it directly or indirectly affects T lymphocytes. PDE1 is distributed in epithelial cells, macrophages, and lymphocytes, and all of these cells can affect allergic lung inflammation, although the exact mechanism is currently unknown. In further studies, we want to decipher the relationship between PDE1 and immune cells through research at the primary cell level.

5. Are other side effects observable upon the systemic application of the inhibitors?

→ Thank you for your helpful comment. According to a previous reference, the systemic application of PDE inhibitors is not known to cause toxicity at the dose used in the present study [1]. In other previous studies, a minimum of 10 mg/kg dose of Vinp was typically used in mice [2, 3]. Thus, we decided to use 10 mg/kg of Vinp, and this dose did not show any side effects in the study.

References

1. Chemical Information Review Document for Vinpocetine [CAS No. 42971-09-5]. National Toxicology Program. U.S. Department of Health and Human Services 2013. https://ntp.niehs.nih.gov/ntp/htdocs/chem_background/exsumpdf/vinpocetine091613_508.pdf

2. Kim NJ, Baek JH, Lee J, Kim H, Song JK, Chun KH. A PDE1 inhibitor reduces adipogenesis in mice via regulation of lipolysis and adipogenic cell signaling. Exp Mol Med. 2019;51(1):1-15.http://doi.org/10.1038/s12276-018-0198-7

3. Ruiz-Miyazawa KW, Pinho-Ribeiro FA, Zarpelon AC, Staurengo-Ferrari L, Silva RL, Alves-Filho JC, et al. Vinpocetine reduces lipopolysaccharide-induced inflammatory pain and neutrophil recruitment in mice by targeting oxidative stress, cytokines and NF-kappaB. Chem Biol Interact. 2015;237:9-17.http://doi.org/10.1016/j.cbi.2015.05.007

Minor:

1. Please provide information of the solvent of IBMX, Vinp, or Dex and the total volume injected.

→ Thank you for your helpful comment. We included the information on the solvent of IBMX, Vinp, or Dex and the total volume injected.

Page 6, Lines 113-115

Stock solutions of IBMX, Vinp, and Dex were prepared at 3.5 mg/ml in DMSO. Before the intraperitoneal (i.p.) injection to the mice, IBMX, Vinp, and Dex were diluted to 1 mg/ml with normal saline.

Page 7, Line 138-140

The administration dosage was 10 mg/kg (the injection volume was 0.2 ml per mouse). In the OVA group, 0.2 ml of the vehicle was injected (i.p.) 1 h before 5% OVA exposure (Fig 1A).

2. Please add treatments in the schedule for inducing asthma (Fig. 1A)

→ Thank you for your helpful comment. Accordingly, we revised Fig 1A to add treatments in the schedule for inducing asthma.

3. Please specify the “airway resistance” measured with a double chamber plethysmograph.

→ Thank you for your helpful comment. Accordingly, we specified the meanings of “airway resistance” and “tidal volume” and described how they were measured.

Pages 7 and 8, Lines 143-154

Airway resistance is the resistance of the respiratory tract to airflow during inhalation and exhalation. Tidal volume is the lung volume representing the normal volume of air displaced between normal inhalation and exhalation when extra effort is not applied. Airway resistance and tidal volume can be measured by plethysmography. ~~~~~ Measured airway resistance and tidal volume were automatically calculated by FinePointe software (Data Sciences International) and expressed as a ratio from the value at 0 mg/ml (n = 9).

4. Please specify how total protein concentrations for OVA specific IgE and Cytokines were calculated.

→ Thank you for your helpful comment. The protein concentration was calculated using a standard curve of manufacturer’s IgE standard solution that was measured at 450 nm. We described the procedure for calculating the protein concentration of OVA-specific IgE and cytokines in the Methods.

Page 10, Lines 201-204

Absorbance was measured using a FlexStation3 Microplate Reader (Molecular Devices, Sunnyvale, CA, USA) at 450 nm according to the manufacturer's protocol. The concentration was determined using a standard curve of manufacturer’s IgE standard solution.

Page 10, Lines 209

Absorbance was measured as described above.

5. Please specify how relative mRNA expression was determined.

→ Thank you for your helpful comment. We detailed the procedure for calculating the relative mRNA expression in the Methods.

Page 11, Lines 2�6

CFX Manager software was used to automatically calculate the cycle quantification value at which samples amplified at a high enough value to be detected (ΔCt values). Each ΔCt value was normalized against GAPDH, used as a housekeeping gene. Transcript expression was determined relative to the control group. The RT-qPCR analysis was performed in three independent experiments (n = 6�8 per group)

6. Please enlarge y-axis labeling of almost all graphs.

→ Thank you for your helpful comment. Accordingly, we enlarged the y-axis labeling of all figures.

---

## [Decision Letter · Decision Letter 1]

17 Feb 2021

PONE-D-20-32932R1

Vinpocetine alleviates lung inflammation via macrophage inflammatory protein-1β inhibition in an ovalbumin-induced allergic asthma model

PLOS ONE

Dear Dr. Lee,

Thank you for submitting your manuscript to PLOS ONE. After careful consideration, we feel that it has merit but does not fully meet PLOS ONE’s publication criteria as it currently stands. Therefore, we invite you to submit a revised version of the manuscript that addresses the points raised during the review process.

Many thanks for having adressed almost all of the comments raised by the two reviewers. However, we feel that in view of your statement that your study "focused on the contribution of PDE1 to asthma and its potential as a therapeutic target", the impact of your paper could be clearly improved if you not only discuss your data on Vinp but also in view of other studies using different PDE inhibitors in asthma models. Please revise the discussion accordingly. I will assess your revised manuscript and come to a decision without involving the reviewers anymore.

We look forward to receiving your revised manuscript.

Kind regards,

Heinz Fehrenbach

Academic Editor

PLOS ONE

Reviewers' comments:

Reviewer's Responses to Questions

**Comments to the Author**

1. If the authors have adequately addressed your comments raised in a previous round of review and you feel that this manuscript is now acceptable for publication, you may indicate that here to bypass the “Comments to the Author” section, enter your conflict of interest statement in the “Confidential to Editor” section, and submit your "Accept" recommendation.

Reviewer #1: All comments have been addressed

Reviewer #2: (No Response)

2. Is the manuscript technically sound, and do the data support the conclusions?

Reviewer #1: Yes

Reviewer #2: Yes

3. Has the statistical analysis been performed appropriately and rigorously? 

Reviewer #1: Yes

Reviewer #2: Yes

4. Have the authors made all data underlying the findings in their manuscript fully available?

Reviewer #1: Yes

Reviewer #2: Yes

5. Is the manuscript presented in an intelligible fashion and written in standard English?

Reviewer #1: Yes

Reviewer #2: Yes

6. Review Comments to the Author

Reviewer #1: NA

Reviewer #2: The present manuscript adressed all my comments. Especially the material and method part is clearly improved.

However, the discussion part still strongly focuses on the own results, here a comparison to other PDE inhibitors would be desirable.

7. PLOS authors have the option to publish the peer review history of their article (what does this mean?). If published, this will include your full peer review and any attached files.

Reviewer #1: No

Reviewer #2: No

---

## [Author Response · Author response to Decision Letter 1]

11 Mar 2021

PLOS ONE

Manuscript number: PONE-D-20-32932

Title: Vinpocetine alleviates lung inflammation via macrophage inflammatory protein-1β inhibition in an ovalbumin-induced allergic asthma model

Revision date: March 10, 2021

We would like to thank you and the reviewers for the comments on our manuscript, which is entitled as “Vinpocetine alleviates lung inflammation via macrophage inflammatory protein-1β inhibition in an ovalbumin-induced allergic asthma model” We revised the manuscript according to the reviewer’s helpful comments. We made red color revised part. We hope that this revised manuscript can be suitable for PLOS ONE’s publication criteria.

Review Comments to the Author

Reviewer #1: NA

Reviewer #2: The present manuscript adressed all my comments. Especially the material and method part is clearly improved. However, the discussion part still strongly focuses on the own results, here a comparison to other PDE inhibitors would be desirable.

Author’s response

Thank you for your helpful comment. We revised the discussion part to discuss not only PDE 1 based on our data but also another PDE subtype using other previous studies using different PDE inhibitors in asthma models

[Line 332-336]

It was known that PDE inhibitors have effect on respiratory disease such as asthma and chronic obstruction pulmonary disease [22, 39]. In particular, PDE 3, 4, and 7 inhibitors are known to be effective in mouse models of asthma from previous studies [22]. However, studies on asthma using the PDE 1 inhibitor in allergic asthma mice model have not been studied yet.

[Line 346-351]

Previous research reported that PDE4 inhibition was shown to have an effect on the attenuation of airway resistance in asthma mice model [43]. PDE3 inhibitor and PDE1/4 dual inhibitor, has been reported to cause the inhibition of the ovalbumin‐induced bronchoconstriction [44]. These previous studies supported that PDE inhibition can alleviated the increased airway resistance. Through this study, we showed that PDE 1 inhibitor can ameliorated the airway resistance increased in asthma mice model.

[Line 357-364]

It was reported that PDE 4 inhibitor reduced the number of eosinophils, neutrophils, and lymphocytes in BALF [47]. …………….………..The effect on inflammatory cells was similar to previous reports using PDE 4 inhibitors.

[Line 378-380]

In previous research, among the PDE inhibitors, PDE 4 inhibitors were reported to affect the inhibition of inflammatory cytokines of BALF (IL-4, 5 and 13) [43].

[Line 417-421]

Through previous studies, it was already demonstrated that PDE4A, 4C, and 4D expression level was increased in allergic pulmonary inflammatory conditions [57, 58]. These research showed the PDE4 was associated with asthmatic conditions. In this study, PDE1A, 1B, and 1C mRNA expression levels and protein expression levels in the lungs were all increased. These results show that PDE1 expression is also associated with asthma.

[References]

22. Page CP. Phosphodiesterase inhibitors for the treatment of asthma and chronic obstructive pulmonary disease. Int Arch Allergy Immunol. 2014;165(3):152-64.http://doi.org/10.1159/000368800 PMID: 25532037

39. Kwak HJ, Nam JY, Song JS, No Z, Yang SD, Cheon HG. Discovery of a novel orally active PDE-4 inhibitor effective in an ovalbumin-induced asthma murine model. Eur J Pharmacol. 2012;685(1):141-8.https://doi.org/10.1016/j.ejphar.2012.04.016 PMID: 22554769

43. Kim SW, Kim JH, Park CK, Kim TJ, Lee SY, Kim YK, et al. Effect of roflumilast on airway remodelling in a murine model of chronic asthma. Clin Exp Allergy. 2016;46(5):754-63.https://doi.org/10.1111/cea.12670 PMID: 26542330

44. Myou, Fujimura, Kurashima, Tachibana, Hirose, Nakao. Effect of aerosolized administration of KF19514, a phosphodiesterase 4 inhibitor, on bronchial hyperresponsiveness and airway inflammation induced by antigen inhalation in guinea-pigs. Clin Exp Allergy. 2000;30(5):713-8.https://doi.org/10.1046/j.1365-2222.2000.00782.x PMID: 10792364

47. Sun J-g, Deng Y-m, Wu X, Tang H-f, Deng J-f, Chen J-q, et al. Inhibition of phosphodiesterase activity, airway inflammation and hyperresponsiveness by PDE4 inhibitor and glucocorticoid in a murine model of allergic asthma. Life Sci. 2006;79(22):2077-85.https://doi.org/10.1016/j.lfs.2006.07.001 PMID: 16875702

57. Tang H-F, Song Y-H, Chen J-C, Chen J-Q, Wang P. Upregulation of phosphodiesterase-4 in the lung of allergic rats. Am J Respir Crit Care Med. 2005;171(8):823-8.http://doi.org/10.1164/rccm.200406-771OC PMID: 15665325

58. Trian T, Burgess JK, Niimi K, Moir LM, Ge Q, Berger P, et al. β2-agonist induced cAMP is decreased in asthmatic airway smooth muscle due to increased PDE4D. PLoS One. 2011;6(5):e20000.http://doi.org/10.1371/journal.pone.0020000 PMID: 21611147

---

## [Decision Letter · Decision Letter 2]

19 Apr 2021

Vinpocetine alleviates lung inflammation via macrophage inflammatory protein-1β inhibition in an ovalbumin-induced allergic asthma model

PONE-D-20-32932R2

Dear Dr. Lee,

We’re pleased to inform you that your manuscript has been judged scientifically suitable for publication and will be formally accepted for publication once it meets all outstanding technical requirements.

Kind regards,

Saba Al Heialy

Academic Editor

PLOS ONE

Additional Editor Comments (optional):

Reviewers' comments:

Reviewer's Responses to Questions

**Comments to the Author**

1. If the authors have adequately addressed your comments raised in a previous round of review and you feel that this manuscript is now acceptable for publication, you may indicate that here to bypass the “Comments to the Author” section, enter your conflict of interest statement in the “Confidential to Editor” section, and submit your "Accept" recommendation.

Reviewer #2: All comments have been addressed

Reviewer #3: All comments have been addressed

2. Is the manuscript technically sound, and do the data support the conclusions?

Reviewer #2: Yes

Reviewer #3: Yes

3. Has the statistical analysis been performed appropriately and rigorously? 

Reviewer #2: N/A

Reviewer #3: Yes

4. Have the authors made all data underlying the findings in their manuscript fully available?

Reviewer #2: Yes

Reviewer #3: Yes

5. Is the manuscript presented in an intelligible fashion and written in standard English?

Reviewer #2: Yes

Reviewer #3: Yes

6. Review Comments to the Author

Reviewer #2: The authors adressed all my recommendations and improved the discussion. I appreaciate the efforts the authors have done.

Reviewer #3: After analyzing both rounds of the manuscript reviews , I can see that the authors addresses most of the reviewers as well as the editors concerns in a fair way.

Results

Regarding the quantitative analysis of histopathological changes, I think the editor made an important point regarding the specificity of the staining. I have 2 suggestions to improve the quality of Figure 2 and I wish the authors will consider those suggestions before publication:

Figure 2A, OVI+ IBMX , (10X) might need to be changes as there is some aggregate of cells that might mimic inflammatory cells and affect the interpretation.

Figure 2C, the last picture also might need to be replaced by better picture as I can see strong background that might give wrong impression about positive cells.

Discussion

In review of the reviews comments as well as the author responses , the discussion was fairly improved to address the literature in the field in addition to the authors own results

7. PLOS authors have the option to publish the peer review history of their article (what does this mean?). If published, this will include your full peer review and any attached files.

Reviewer #2: No

Reviewer #3: **Yes: **Ibrahim Yaseen Hachim

---

## [Editor Report · Acceptance letter]

21 Apr 2021

PONE-D-20-32932R2 

Vinpocetine alleviates lung inflammation via macrophage inflammatory protein-1β inhibition in an ovalbumin-induced allergic asthma model 

Dear Dr. Lee:

I'm pleased to inform you that your manuscript has been deemed suitable for publication in PLOS ONE. Congratulations! Your manuscript is now with our production department. 

Kind regards, 

on behalf of

Dr. Saba Al Heialy 

Academic Editor

PLOS ONE